# Language-assisted Feature Representation and Lightweight Active Learning For On-the-Fly Category Discovery

**Anwesha Banerjee**                                              *anweshabaner@iisc.ac.in*
*Department of Electrical Engineering Indian Institute of Science, Bengaluru*

**Soma Biswas**                                                  *somabiswas@iisc.ac.in*
*Department of Electrical Engineering Indian Institute of Science, Bengaluru*

**Reviewed on OpenReview:** *https://openreview.net/forum?id=ZihFoM8KOj*

## Abstract

Contemporary deep learning models are very successful in recognizing predetermined categories, but often struggle when confronted with novel ones, constraining their utility in the real world. Identifying this research gap, *On-the-fly Category Discovery* aims to enable machine learning systems trained on closed labeled datasets to promptly discern between novel and familiar categories of the test-images encountered in an online manner (one image at a time), along with clustering the different new classes as and when they are encountered. To address this challenging task, we propose **SynC**, a pragmatic yet robust framework that capitalizes on the presence of category names within the labeled datasets and the powerful knowledge-base of Large Language Models to obtain unique feature representations for each class. It also dynamically updates the classifiers of both the seen and novel classes for improved class discriminability. An extended variant, **SynC-AL** incorporates a lightweight active learning module to mitigate errors during inference, for long-term model deployment. Extensive evaluation show that SynC and SynC-AL achieve state-of-the-art performance across a spectrum of classification datasets.

## 1 Introduction

Deep learning models have achieved remarkable success in visual recognition tasks, often surpassing human performance. However, these models typically rely on closed category sets and extensive annotated datasets, limiting their ability to recognize novel categories. This presents a challenge in real-world, where encountering new categories is inevitable. The abundance of unlabeled data increases the need for models that can generalize to new categories and identify similarities without extensive annotation. Generalized Category Discovery (GCD) (S. Vaze, 2022) aims to classify samples in a query set containing both known and unknown categories, based on a labeled support set of known categories. The challenge lies in accurately identifying new, unseen classes that share semantic similarities with the known ones. Still, the GCD task relies on (i) access to the data from test categories during training, which limits their ability to handle truly novel classes, and (ii) batch-wise processing during inference, which is not desirable for real-time applications. To address the limitations of GCD, (Du et al., 2023) proposed a more challenging and realistic task, namely *On-the-fly Category Discovery (OCD)* (Figure 1). OCD utilises a closed-labeled set for training without requiring access to novel class data. Additionally, it performs inference for each query sample in an online manner, and obviates the requirement of collecting batches of test data. This shift from the standard paradigm creates a agile and adaptable classification framework, better suited for real-world applications.

Here, we propose a novel framework termed **SynC** *(**Syn***cing image features and Language-assisted representations with **C***lassifier Update)*, which is designed to overcome these challenges. Since the model encounters images from novel classes during inference, it is important that the model learns to map input training images to a semantically meaningful latent feature space, which is unique for each class without any apriori

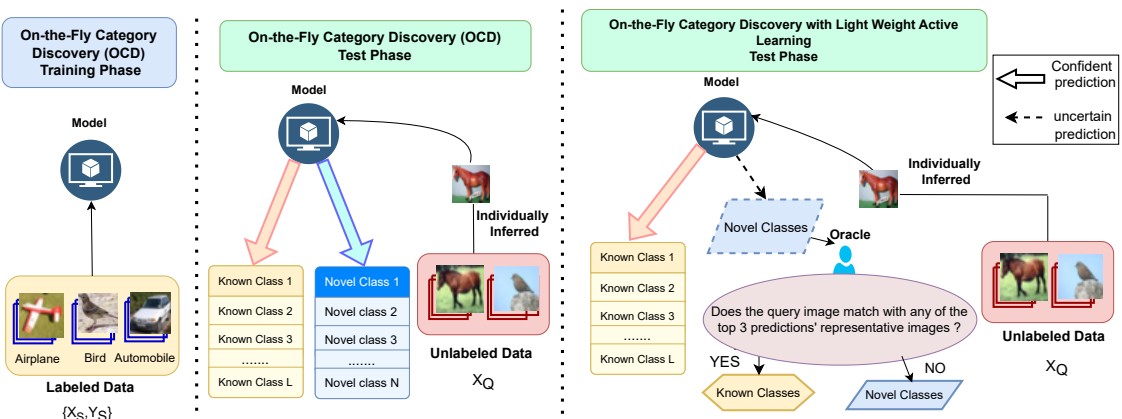

Figure 1: Different stages of OCD: Left: Training, Middle: Testing and Right: Testing with Active Learning. We propose SynC and SynC-Al for handling both the testing scenarios.

knowledge about these classes. Towards this goal, as a pre-processing step, we utilize category names from the labeled dataset to generate descriptions using the powerful knowledge-base of Large Language Models (LLMs). By providing prompts specific to each category, we obtain comprehensive textual representations that encode the semantic essence of each category. We use two tailored supervised contrastive loss functions to align the image embeddings with the semantic information embedded within their class-specific textual representations and class-wise prototypes. During inference, SynC categorizes the samples sequentially into previously encountered or novel classes by utilizing a class-wise adaptive threshold. By utilizing the nearest mean class representatives as the classifiers, learned associations between visual and textual modalities are leveraged, thereby enhancing classification accuracy. During test-time, SynC utilizes the stored statistics of the classes encountered so far to periodically update all the classifiers simultaneously, thereby further improving their discriminability.

Furthermore, we propose an extended variant, **SynC-AL**, which incorporates *lightweight active learning*, allowing for effective human intervention to correct errors that could otherwise propagate through the system. Even with minimal human involvement, SynC-AL can help to further boost the performance of the framework. Experimental results demonstrate that the proposed framework achieves state-of-the-art performance across three widely used coarse-grained and four fine-grained classification datasets. **The contributions of the work are as follows:**

1) We propose a novel SynC framework which effectively utilizes language supervision to address the challenging OCD task.We show that language supervision can be successfully leveraged for the OCD task, even when the class names of the novel categories are unknown.

2) We propose a test-time classifier update during inference to allow knowledge transfer between old and novel classes for improved discriminability.

3) To the best of our knowledge, this is the first work to propose a strong baseline (SynC-AL) for On-the-fly-Active Category Discovery (OACD) setting.

4) We establish state-of-the-art results for this challenging task on several benchmarks.

## 2 Related Work

Here, we provide pointers to a few related work on the different concepts used in this work.

**Generalized Category Discovery (GCD)**: GCD (S. Vaze, 2022) involves classifying unlabeled data containing instances from both previously seen classes and entirely novel classes. Here, both unsupervised and supervised contrastive losses (C. Ting, 2020) are employed to fine-tune a Vision Transformer (ViT) (A. Kolesnikov & Zhai, 2021), that was pre-trained using DINO (C. Mathilde, 2021). This enhances the similarity between feature representations of various views of the same image and between different images of the same class. PromptCAL (Z. Sheng, 2023) introduced prompts to provide valuable semantic

supervision signal for the GCD task while AMEND (Banerjee et al., 2024) offers similar performance while using neighboorhood information. In P. Nan (2023), conceptual contrastive learning is utilized by the authors, which considers the relationship between instances and significantly improves clustering accuracy. SPTNet (W. Hongjun, 2024) proposes a two-stage adaptation approach that optimizes both model and data parameters by accounting for the spatial properties of image data. Tingzhang et al. in L. Tingzhang (2024) propose a dual-context approach that enhances feature learning by integrating instance-level and cluster-level contextual information, thereby improving the identification and classification of categories in unlabeled datasets. Keon-Hee et al. in P. Keon-Hee (2024) introduce a novel problem setting "Online Continuous Generalized Category Discovery". They also propose a framework that enables models to continuously learn and adapt to new categories in streaming data using energy-guided discovery, variance-based feature augmentation, and contrastive loss to mitigate forgetting and enhance category separation. *On-the-fly Category Discovery addressed in this work emphasizes real-time adaptation with instantaneous category discovery, requiring each sample to be inferred independently, whereas Online Continuous Generalized Category Discovery allows multiple test-time samples and often incorporates memory buffers or replay strategies.*

**On-the-fly Category Discovery:** The authors of (Du et al., 2023) introduced the practical OCD task to enable trained models to instantaneously recognize and classify novel categories, akin to human cognition. They propose a hash coding-based recognition model and a Sign-Magnitude dIsentangLEment (SMILE) architecture to mitigate intra-category variance, facilitating inductive learning and streaming inference for rapid adaptation to new information. Haiyang et al in Z. Haiyang (2024) introduce a Prototypical Hash Encoding (PHE) framework that enhances online discovery of new categories in streaming data by generating multiple prototypes per category to capture intra-category diversity and employing discriminative category encoding to improve hash code discrimination, effectively addressing sensitivity issues in fine-grained class discovery. *Both the works relies of hash codes, hence they faces challenges related to the predefined length of hash codes for category descriptors, which is addressed in our work.*

**Self-supervision:** SimCLR (C. Ting, 2020) proposes a framework for contrastive learning of visual representations, which learns effective features from unlabeled data by maximizing agreement between augmented views of the same image while minimizing agreement between views of different images. Supervised contrastive learning (K. Prannay, 2020) extends these methods by incorporating labeled data, where positive pairs consist of samples from the same class and negative pairs are from different classes. *Our work leverages both supervised learning and contrastive learning to learn effective representations.*

**Zero-shot learning:** CHiLS(Novack et al., 2023) improves CLIPs zero-shot accuracy on coarse classes by expanding each superclass into a set of finer hierarchical subclasses, sourced from taxonomies or generated via GPT-3, and then mapping subclass predictions back to their original superclasses through a simple reweighting of superclass probabilities. This work demonstrates that leveraging predefined hierarchical label structures can substantially boost zero-shot accuracy within a fixed taxonomy. However, it operates under the assumption that the full class hierarchy is known in advance and does not address settings involving truly novel categories.The authors of P. Sarah (2023) use large language models to automatically generate customized, class-specific natural language prompts, conditioned on class names, attributes, and contextual cues, instead of relying on fixed template phrases. These tailored prompts, when used with vision-language models such as CLIP, lead to significantly improved zero-shot classification accuracy by capturing richer semantic nuances for each category. S3A (Zhang et al., 2024) addresses a more realistic zero-shot setting in which neither labeled examples nor ideal vocabularies are available. It proposes a ClusterVotePromptRealign pipeline, which extracts structural semantics from unlabeled images by clustering them, voting on candidate labels from a broad vocabulary, refining prompts with a language model, and realigning prototypes for pseudo-supervision. The model then self-trains the CLIP image encoder using both individual and structural semantic alignment in a teacherstudent framework. However, S3A assumes access to all unlabeled data upfront, whereas in the OCD setting, samples arrive in a streaming manner and inference must be performed on-the-fly, making the two settings fundamentally different.

**Textual supervision:** In this paper (Gao et al., 2022), the authors introduce KeyClass, a weakly supervised framework that automatically derives interpretable labeling functions from classlabel descrip-

tions and combines them via data programming to train text classifiers without any humanannotated documents . Evaluated on both benchmark datasets and ICD-9 code assignment for MIMIC-III notes, KeyClass achieves accuracy on par with fully supervised models, significantly reducing the need for costly manual labeling .

**Active Learning:** Traditionally, active learning is performed in a closed-world setting, where the labeled and unlabeled data contains the same set of classes to maximize the performance of a model with a limited labeling budget (Settles, 2009). The widely used pool-based methods (Z. Xueying, 2021) can be broadly categorized into two groups: (i) diversity-based methods (S. Ozan, 2018; A. Jordan T, 2020) choose samples that effectively represent the entire dataset; (ii) uncertainty-based methods (R. Dan, 2006) prioritize selecting samples that exhibit high levels of predictive uncertainty. Hybrid methods (A. Sharat, 2020; H. Sheng-Jun, 2010) that combines both approaches for better performance have also been proposed. To overcome the issues of imbalanced classification performance and inconsistent confidence between old and new classes, inspired by active learning principles, Active Generalized Category Discovery (AGCD) (M. Shijie, 2024) is proposed. AGCD aims to improve GCD performance by strategically selecting valuable samples for labeling from an oracle, using an adaptive sampling strategy that considers novelty, informativeness, and diversity. *Here, we propose an active learning strategy for the very challenging and realistic OCD setting.*

## 3 Proposed Framework

In this work, we address the challenging OCD task, where the focus is on enabling machine learning models to adeptly identify previously unseen categories during test-time in an online manner. The central concept revolves around training conducted on a closed labeled dataset, subsequently requiring the model's capability to discern novel and familiar categories instantaneously. The model is first trained on a support set $D_S = \{(x_i, y_i)\}_{i=1}^{S} \in X_S \times Y_S$, encompassing $S$ image samples $x_i$ paired with corresponding labels $y_i$. During inference, the model encounters query images from $D_Q = \{x_i\}_{i=1}^{Q} \in X_Q$ as they arrive one at a time in an online manner. The test set $D_Q$ comprises both known (seen during training) and novel categories, i.e. the set of training labels is a subset of the query labels ($Y_S \subset Y_Q$). The task is to assign the query samples to either previously encountered classes or novel, unseen categories.

It is to be noted that, zero-shot learning assumes a fixed set of target classes are known a priori. ZSL also uses external semantics (like attributes or text prompts) to label samples from unseen classes, whereas in case of On-the-fly Category Discovery, the model dynamically discovers new categories relying only on semantic information from the already seen classes.

The proposed framework SynC has two modules each, in both the train and test stages: The main modules of the training stage using examples from known classes are: 1) Generation of unique LLM-assisted class representations and 2) Syncing image features with unique textual description and class prototypes. The two modules of the test stage are: 1) Adaptive threshold for novel class discovery and 2) Test-time update of the classifiers. In addition, if human involvement is allowed during inference, we present another variant of our framework, SynC-AL, where a *lightweight active learning module* is incorporated to mitigate inherent inaccuracies from propagating further. Now, we describe the different modules in detail.

### 3.1 SynC Training Phase

Given training samples from known classes, the goal of SynC is to learn a mapping from the image to a semantic latent space such that the images of novel classes seen during testing will also be embedded in a semantically meaningful manner. We achieve this using the following two modules.

#### 3.1.1 Generation of unique LLM-assisted class representations

We aim to learn a mapping from the images to a semantically meaningful latent space, such that images belonging to the same class (known or novel) automatically gets mapped to a unique location in the latent space. The dataset $D_S$ contains labels $y_i$ corresponding to each sample $x_i \in X_S$, which are used to obtain its

category name $c_l$. To obtain the unique text representation of each class, we utilize the powerful knowledge-base of the pre-trained Large Language Models (LLM). First, the LLM is supplied with prompts $(v_1, v_2, .., v_d)$ specific to each category name, to elicit distinctive descriptions for each category. Specifically, each prompt $v_i$ augmented with the category name $c_l$ is given as input to the LLM, which generates a textual description $T_i$, i.e. $h^{llm} : \{v_i + c_l\} \mapsto T_i$. For example, given the category name Black-footed Albatross, the prompts used for obtaining descriptions of this category are *'Describe the bird, Black-footed Albatross:',' What are the identifying characteristics of the bird, Black-footed Albatross?'* and *'Describe what the bird Black-footed Albatross looks like'* (P. Sarah, 2023). The details of all the prompts used are provided in Table 12 in Appendix A. Thus, for each category, $d$ textual descriptions $(T_1, T_2, .., T_d)$ are acquired for $d$ LLM prompts $(v_1, v_2, .., v_d)$. Subsequently, Sentence-BERT (N, 2019) is employed to extract text embeddings from these descriptions, from which the mean text embedding is computed which is taken as the unique textual representation $t_l$ for the class $c_l$ as seen in Figure 2.

### 3.1.2 Syncing image features with unique textual description and class prototypes

The goal is to train a model on a set of classes, such that it (i) can discern whether a test (query) image belongs to one of the seen classes or a novel class, (ii) is able to group data from distinct novel classes separately and (iii) does not require any knowledge of the number of novel classes that will be encountered. We propose to leverage class-name-based text embeddings and the class-specific image prototypes of the training data, since they are unique for each class, and automatically leads to the desired discrimination in the semantically meaningful latent space. Thus, the goal is to learn the mapping of the images (and their variations) to the unique, semantically meaningful textual descriptions learnt in the previous module, on the assumption that it will subsequently generalize and map images from novel classes to their respective unique locations in the latent space during testing. Specifically, given an image $x_i$, the $l_2$ normalized feature $z_i$ is obtained by using feature extraction backbone $f$ i.e. $z_i = f(x_i)$. During training, we first utilize supervised contrastive loss to bring all images of the same class closer for improved discriminability. Specifically, for $x_i$, its augmentation and all samples within the mini-batch that belongs to the same class as that of $x_i$ are considered as positive samples. The supervised contrastive loss (K. Prannay, 2020) can be written as:

$$\mathcal{L}_i^s = -\frac{1}{|M(i)|} \sum_{q \in M(i)} \log \frac{\exp(z_i \cdot z_q/\alpha)}{\sum_n 1_{[n \neq i]} \exp(z_i \cdot z_n/\alpha)} \tag{1}$$

where $M(i)$ denotes a set of indices of the augmentation and all the other labeled samples belonging to the same class as that of $x_i$ in the mini-batch $B$. $\alpha$ is the temperature used for scaling. To align the images of the same class with their unique textual description, we utilize the textual representation $t_l$ as an adjunct view for the anchor image $x_i$. This association enhances the model's capacity to comprehend the intrinsic characteristics and semantic attributes of the depicted category. The **textual supervised contrastive loss** is given as:

$$\mathcal{L}_i^t = -\log \frac{\exp(z_i \cdot t_l/\alpha)}{\sum_n 1_{[n \neq i]} \exp(z_i \cdot t_n/\alpha)} \tag{2}$$

This alignment enhances the model's ability to establish meaningful connections between the visual and textual modalities.

Furthermore, to enhance the clustering performance of the model, we propose to include the prototypical class representative as another positive sample within this framework of supervised contrastive loss. Specifically, for each class $c_l$, where $c_l \in Y_S$, a class-specific image prototype $p^l$ is computed by taking the mean of all $l_2$ normalized image features belonging to the class $c_l$. To facilitate the learning process and ensure that the image embeddings are more closely aligned with the corresponding class prototype, we utilize the class prototype $p^l$ as another positive pair. and the **prototypical supervised contrastive loss** is given as:

$$\mathcal{L}_i^p = -\log \frac{\exp(z_i \cdot p^l/\alpha)}{\sum_n 1_{[n \neq i]} \exp(z_i \cdot p^n/\alpha)} \tag{3}$$

This improves the overall clustering and classification performance. Thus, the unified training objective is:

$$\mathcal{L}_{\text{total}} = \mathcal{L}^s + \mathcal{L}^t + \mathcal{L}^p \tag{4}$$

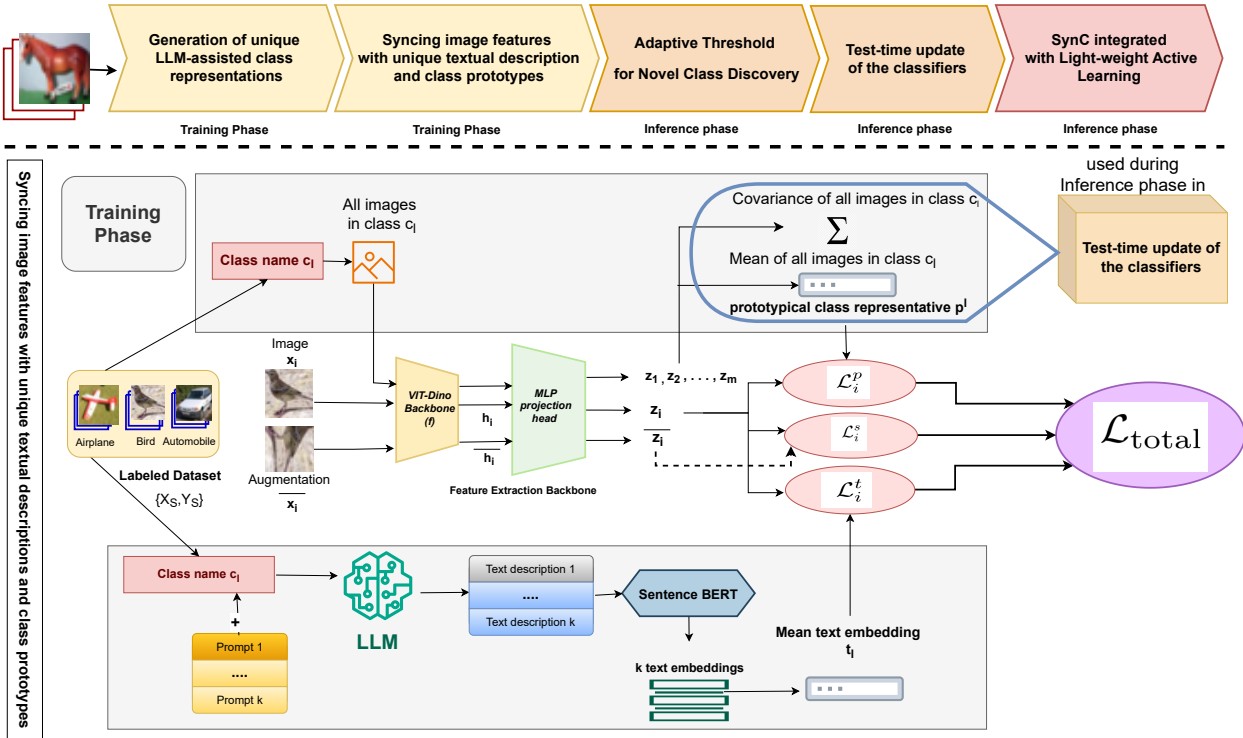

Figure 2: Overall structure of our framework SynC and SynC-AL. The second module 'Syncing image features with unique textual description and class prototypes' is demonstrated in detail here. In the training phase, each image is passed through a ViT backbone and MLP projection head to extract image features. For each class $c_l$, multiple prompts are used to generate k textual embeddings whose mean defines a unique LLM-based class representation $t_l$. Simultaneously, the mean of the image features for all samples in class $c_l$ forms the prototypical class representative $p_l$. These are jointly used to align image and text features using the loss $\mathcal{L}_{total}$. During inference, use of adaptive thresholds helps discovery of instances from novel class, while light-weight active learning and test-time updates refine the classifier progressively.

Jointly minimizing $\mathcal{L}_{total}$ ensures that the learned embeddings simultaneously capture visual consistency within classes, align closely with semantic textual descriptions, and cluster effectively around class prototypes.

### 3.2 SynC Inference Phase

During inference, the model trained as described above encounters individual samples sequentially. The task is to categorize them into previously encountered classes or designate them as originating from a novel, unseen class. As novel classes are discovered, in order to improve discriminability among all the classes, the classifiers are updated periodically using the stored statistics of all the classes encountered so far.

#### 3.2.1 Adaptive Threshold for Novel Class Discovery

Given the integration of class prototypes as positive samples within the supervised contrastive loss framework, we propose to employ class prototypes as the classifier during testing. Given there are $L$ labeled classes during training, cosine similarity between the dictionary of prototypical class representatives $P = \{p^1, p^2, ..., p^L\}$ of previously seen classes and the $l_2$ normalized feature $z_i$ of the current image $x_i$ is calculated as $sim_P = <z_i, P>$ To account for the variability among different classes, i.e. some classes may be more clustered compared to others which are more spread out, we utilize statistical modeling for setting adaptive thresholds for different classes, thus making it more adaptable to the data distribution of each class. Specifically, for each known class $c$ where $c = \{1, 2, ..L\}$, the mean, $M_c$ and standard deviation $\epsilon_c$ of the cosine similarities between $l_2$ normalized image features and their corresponding prototypical class representatives are computed. The

class specific threshold $\tau_c$ for each class $c$ is then determined as follows:

$$\tau_c = M_c - g * \epsilon_c \tag{5}$$

where $g$ is a hyperparameter. This ensures that the threshold is higher for the well clustered classes, and smaller for the spread out ones. The difference between the cosine similarity of the current image feature and the prototypical class representative $p^c$ is calculated and the sample is assigned to class $c$ if the similarity exceeds the class-specific threshold. Otherwise, the sample is presumed to belong to a novel, unseen class. In this case, in the absence of any other information about the newly discovered category (like class name), we propose to store the $l_2$ normalized feature $z_i$ corresponding to the image sample as the classifier for this novel category. Hence, the first novel class representative is denoted as $Z^1 = z_i$. Consequently, the dictionary comprising of the prototypical class representatives of the classes seen so far is represented as $(p^1, p^2, .., p^L, Z^1)$.

Since we have limited access to samples from novel classes, we assign the threshold for an incoming novel class to be the same as the class-specific threshold of the known class that it is most similar to, based on their latent feature representations. This approach assumes that semantically similar classes will exhibit similar variations in their feature distributions.

### 3.2.2 Test-time update of the classifiers

During the testing phase, for the samples designated as belonging to novel classes, the image feature is taken as the classifier, in the absence of their actual class names. We make two observations: (i) Depending upon the images encountered, these features may not be very good representatives of the corresponding class; (ii) This also leads to loss of discriminability when the old and new classes are considered together. To overcome these challenges, we propose to update the classifiers periodically, thereby sharing knowledge between all classes encountered so far.

Suppose, at a certain instant, $m$ number of novel classes ($m \leq N$, where $N$ is the total number of novel categories present in the query set $D_Q$) have been identified so far. The dictionary of classifiers at this point is given by $(p^1, p^2, .., p^L, Z^1, Z^2, .., Z^m)$. For updating the classifiers, one straightforward solution is to store few training images of the classes encountered so far, but this may have storage/privacy concerns. Instead, we utilize the means $\mu_c$ and covariances $\sum_c$ for each class $c$, the underlying assumption is that each class can be modeled as a Gaussian distribution $\mathcal{N}(\mu_c, \sum_c)$. For the seen (training) classes, we can properly estimate the mean and covariances from the training data. Since we do not have access to many samples from the novel categories, we use the classifier feature $Z^j$ where $j\epsilon\{1, 2, ..., m\}$ as the mean feature for that category and the covariance of the most similar class (based on their latent features), assuming that semantically similar classes will exhibit similar variations. Details of experiments supporting the claim that "semantically similar classes will exhibit similar variations", is present in Appendix A.4.
Finally, features $\hat{F}_c = \hat{z}_c^h, h = 1, \ldots, H$ are sampled from the distribution $\mathcal{N}(\mu_c, \sum_c)$ for each class $c \in 1, 2, ..., L+m$, where $H$ represents the number of generated features per class. Subsequently, the standard cross-entropy loss is used to adjust the classifiers by feeding these features into the classifier head, which is a single-layer perceptron with a dictionary of classifiers as its weights. This classifier adjustment results in better separation of the classifiers with respect to each other and is performed periodically after each new set of $m$ novel classes is discovered.

## 4 SynC-AL: SynC integrated with Light-weight Active Learning

In the On-the-fly Category Discovery (OCD) task, models must continuously adapt to novel classes encountered during inference. However, this real-time adaptability poses a fundamental challenge: as the model incrementally processes individual samples, prediction errors (eg. misclassification of a seen class instance as novel) can quickly accumulate, leading to compounding inaccuracies over time as seen in Figure 4. Although the adaptive threshold strategy in SynC mitigates some errors by dynamically adjusting decision boundaries, certain ambiguous or borderline samples remain inherently difficult for automated classification. Active Learning (AL) presents an attractive solution by strategically incorporating limited human supervision to resolve these challenging cases. However, traditional AL approaches, which often involve extensive

human involvement, are impractical for real-time or resource-constrained applications. Thus, an effective AL method specifically tailored to the OCD task must be lightweight, efficient, and impose minimal cognitive load on human annotators.

Motivated by these practical constraints, we propose **SynC-AL**, a lightweight active learning extension of SynC. Unlike traditional AL methods that require extensive labeling or domain expertise, SynC-AL leverages simple binary comparisons with class-representative images to efficiently resolve uncertain predictions. Additionally, we introduce a *Confusion Buffer* to minimize redundant human queries, further reducing annotation costs without sacrificing accuracy. Through SynC-AL, we demonstrate that even minimal, intelligently directed human input can significantly enhance long-term performance in dynamic, real-world category discovery scenarios. Now we describe the approach in detail.

During testing, given an image sample $x_i$ with an $l_2$-normalized feature representation $z_i$, its similarity with the prototypical class representatives $(p^1, p^2, ..., p^L, p^{(L+1)} = Z^1, p^{(L+2)} = Z^2, ..., p^{(L+m)} = Z^m)$ of previously seen classes (assuming $m$ novel classes have already been encountered) is computed. If the maximum cosine similarity between $z_i$ and any known class prototype $p^c$ is less than the corresponding class-specific threshold $\tau_c$, i.e.

$$\max_{c \in \{1, ..., L+m\}} \cos(z_i, p^c) < \tau_c, \tag{6}$$

the sample is considered a probable instance of a novel category and is processed by the active learning module. Thresholding the cosine similarity for novel category discovery introduces potential inaccuracies, one common one being some instances of previously seen classes may not achieve a similarity score above their respective threshold $\tau_c$, resulting in their misclassification as a novel class. To mitigate such errors and improve the long-term performance of the model, we introduce a light-weight active learning mechanism in our framework, which we term as SynC-AL.

Let $B_{\text{active}}$ denote the active learning budget, representing the maximum number of queries for which the model can request human feedback. The total human effort depends not only on the budget $B_{\text{active}}$, but also on the complexity of the feedback requested. In our work, when a query image $x_i$ is predicted to belong to a novel class, the active learning module becomes functional. The straightforward human feedback that can be requested is the *class of the input sample*, based on which it can be incorporated in one of the existing categories or classified as a new class. This approach has the following limitations:
i) It requires domain expertise to classify a given image into its class, specifically for fine-grained or specialized datasets like CUB-200-2011, Stanford Cars, Food 101, etc.
ii) It becomes especially harder when the number of classes of interest increases, eg, CUB-200-2011 dataset has 200 classes.

Here, we propose a specialized active learning module, that requires much simpler human feedback. When a query image $x_i$ is predicted to belong to a novel class, our proposed active learning module is engaged to refine the prediction by leveraging the models top-$k$ most likely class predictions. The intuition is that usually, the correct class (if it belongs to one of the already seen classes) lies within the top predictions. In order to facilitate this, for each encountered class $c$, the model maintains a *representative image $r^c$*, encapsulating the distinguishing features of that class. Let $\mathcal{C}_{\text{top-}k}(x_i)$ denote the set of the top-$k$ predicted classes for $x_i$, where each class $c \in \mathcal{C}_{\text{top-}k}(x_i)$ has an associated representative image $r^c$. The active learning module queries an oracle $\mathcal{O}$ to compare the query image $x_i$ with each representative image $r^c$ from the predicted set. The oracle provides a binary response for each comparison as follows:

$$\mathcal{O}(x_i, r^c) = \begin{cases} Yes, & \text{if } x_i \text{ belongs to the same class as } r^c \\ No, & \text{otherwise} \end{cases} \tag{7}$$

If the oracle confirms that $x_i$ matches with any of the top-$k$ predicted classes, the query image is assigned to class $c$. Conversely, if the oracle returns *No* for all $k$ comparisons, the image is classified as belonging to a *novel class*. This active learning process is computationally efficient, as it only requires $k$ *comparisons*, making it a lightweight and practical approach for real-time category discovery. This active learning module does not require significant domain knowledge, like names of fine-grained classes, etc., as the oracle is not tasked with identifying the exact class of the query image. Instead, it only needs to evaluate the similarity

|  | CIFAR10 | CIFAR100 | ImageNet-100 | CUB-200-2011 | Stanford Cars | Food101 | Oxford Pets |
|---|---|---|---|---|---|---|---|
| $|Y_S|$ | 5 | 80 | 50 | 100 | 98 | 51 | 19 |
| $|Y_Q|$ | 10 | 100 | 100 | 200 | 196 | 101 | 38 |
| $|D_S|$ | 12.5K | 20.0K | 31.9K | 1.5K | 2.0K | 19.1K | 0.9K |
| $|D_Q|$ | 37.5K | 30.0K | 95.3K | 4.5K | 6.1K | 56.6K | 2.7K |

Table 1: Details of the datasets used, the standard split in terms of the number of labeled and unlabeled classes ($|Y_S|, |Y_Q|$) and the number of images in the labeled and unlabeled set ($|D_S|, |D_Q|$).

| Components | CIFAR 10 | | | CIFAR 100 | | | Imagenet-100 | | |
|---|---|---|---|---|---|---|---|---|---|
|  | All | Old | New | All | Old | New | All | Old | New |
| SLC (Hartigan, 1975) | 41.54 | **58.29** | 33.29 | 44.36 | 58.98 | 15.10 | 32.92 | **86.55** | 5.22 |
| MLDG (L. Da, 2018) | 44.14 | 38.47 | 46.98 | 50.60 | 60.98 | 29.83 | 30.63 | 72.30 | 9.69 |
| RankStat (H. Kai, 2021) | 42.14 | 49.26 | 38.59 | 35.00 | 44.01 | 16.98 | 31.06 | 73.30 | 9.83 |
| WTA (J. Xuhui, 2021) | 43.12 | 34.52 | 47.42 | 40.83 | 52.89 | 16.72 | 30.84 | 72.92 | 9.68 |
| SMILE (Du et al., 2023) | 49.86 | 39.86 | 54.86 | 51.59 | 61.55 | **31.69** | 33.78 | 74.22 | 13.55 |
| **SynC** | **50.13** | 40.57 | **54.90** | **56.13** | **68.43** | 31.54 | **44.01** | 86.22 | **22.80** |

Table 2: Performance (accuracy %) of the proposed SynC framework on three coarse-grained datasets.

between the query image and the representative images, reducing the cognitive load and making the system more accessible for non-expert users.

**Confusion Buffer**: Given the small active learning budget $B_{\text{active}}$, its careful utilization is crucial. To optimize its use, a repository of confusing samples is maintained to prevent redundant queries. When a sample $x_i$ fails to surpass the class-specific threshold $\tau_c$, but the active learning module confirms that it belongs to one of the previously seen classes, the sample is considered *confusing* and difficult for the model to classify correctly. Formally, if

$$\max_{c \in \{1,...,L\}} \cos(z_i, p^c) < \tau_c, \quad \text{but } \mathcal{O}(x_i, r^c) = Yes \ \& \ r^c \in \text{ top k predicted classes of } \mathcal{C}_{\text{seen}}, \tag{8}$$

where $\mathcal{O}(x_i, r^c)$ represents the oracle-provided ground-truth label and $\mathcal{C}_{\text{seen}}$ is the set of previously seen classes, the sample is stored in a *Confusion Buffer* $\mathcal{M}$, which has a predefined fixed capacity.

To conserve the active learning budget, any new query image $x_q$ from the test set $D_Q$ that doesn't cross the class-specific threshold $\tau_c$ is first cross-checked against the Confusion Buffer. Let $z_q$ be the $l_2$-normalized feature of $x_q$. If the highest cosine similarity between $z_q$ and any stored confusing sample $z_m \in \mathcal{M}$ exceeds a similarity threshold $\tau_c$, i.e. $\max_{z_m \in \mathcal{M}} \cos(z_q, z_m) \geq \tau_c$ the query image is assigned to the corresponding class. This process helps to classify confusing samples without invoking the active learning module. However, if $x_q$ remains dissimilar to all stored samples, i.e., $\max_{z_m \in \mathcal{M}} \cos(z_q, z_m) < \tau_c$, then the active learning module is engaged, provided the query remains within the confines of the active learning budget. By incorporating the Confusion Buffer, unnecessary active learning queries are avoided, thereby optimizing the utilization of the limited budget $B_{\text{active}}$. Detailed Algorithm is given in Appendix A.3.

## 5 Experiments

Extensive experimental evaluations are conducted on three generic and four fine-grained benchmark datasets. The generic datasets encompass CIFAR10, CIFAR100 (Krizhevsky et al., 2009), and ImageNet-100 (T. Yonglong, 2020), and the fine-grained datasets include Stanford Cars (K. Jonathan, 2013), CUB (Welinder et al., 2010), Oxford Pets (P. Omkar M, 2012) and Food 101 (B. Lukas, 2014) (Semantic Shift Benchmark).

These datasets collectively serve as standard benchmarks for assessing the performance and generalization capabilities of our SynC framework for OCD setting and its extended version SynC-AL for OACD setting. From each dataset $D$, a subset of classes is sampled to constitute the set $Y_S$ denoted as the seen classes. Subsequently, 50% of the images associated with each seen class are randomly selected to form the labeled set $D_S$, while the remaining images constitute the unlabeled set $D_Q$. Furthermore, it is assumed that a disjoint validation set is available, where a subset of labels is masked to obtain the unlabeled set. More dataset details are given in Table 1.

| Methods | Oxford Pets | | | Food101 | | | CUB-200-2011 | | | Stanford Cars | | |
|---------|-----|-----|-----|-----|-----|-----|-----|-----|-----|-----|-----|-----|
| | All | Old | New | All | Old | New | All | Old | New | All | Old | New |
| SLC | 35.5 | 41.3 | 33.1 | 20.9 | 48.6 | 6.8 | 28.6 | 44.0 | 20.9 | 14.0 | 23.0 | 9.6 |
| RankStat | 33.2 | 42.3 | 28.4 | 22.3 | 50.7 | 7.8 | 21.2 | 26.9 | 18.4 | 21.2 | 26.9 | 18.4 |
| WTA | 35.2 | 46.3 | 29.3 | 18.2 | 40.5 | 6.1 | 21.9 | 26.9 | 19.4 | 17.1 | 24.4 | 13.6 |
| SMILE | 41.2 | 42.1 | 40.7 | 24.0 | 54.6 | 8.4 | 32.2 | 50.9 | 22.9 | _26.1_ | _46.6_ | 16.2 |
| PHE | _48.3_ | _53.8_ | _45.4_ | _29.1_ | **64.7** | _11.1_ | _36.4_ | **55.8** | _27.0_ | **31.3** | **61.9** | _16.8_ |
| **SynC** | **61.65** | **69.46** | **57.55** | **31.09** | _60.72_ | **15.98** | **45.33** | _54.27_ | **40.85** | 24.60 | 34.80 | **19.55** |

Table 3: Performance (accuracy %) of the proposed SynC framework on four fine-grained datasets.

| Methods | CIFAR 10 | | | CIFAR 100 | | | Imagenet-100 | | |
|---------|-----|-----|-----|-----|-----|-----|-----|-----|-----|
| | All | Old | New | All | Old | New | All | Old | New |
| **Margin-based baseline** | _51.52_ | _20.05_ | _67.25_ | _56.11_ | _72.23_ | _23.86_ | _44.69_ | **90.10** | _21.86_ |
| **Uncertainty-based baseline** | _51.71_ | _20.06_ | **67.54** | _57.80_ | _74.42_ | _24.55_ | _45.17_ | _87.84_ | _23.72_ |
| **SynC-AL** | **52.12** | **39.65** | 58.35 | **57.96** | _72.67_ | **28.56** | **46.00** | _88.04_ | **24.88** |

Table 4: Experimental results showing the baselines of OACD setting and our framework 'Sync-AL' across three generic datasets.

**Evaluation Protocol:** For fair comparison with existing works (Du et al., 2023)(Z. Haiyang, 2024), we employ Strict-Hungarian as our evaluation protocol. The Strict-Hungarian protocol (S. Vaze, 2022) calculates the accuracy of the entire query set at once, thereby mitigating the possibility of clusters being redundantly utilized by both new and old categories. The clustering accuracy (ACC) is calculated by making use of the ground truth labels $y_i$ and the predicted cluster labels $\hat{y}_i$:

$$ACC = \max_{p \in \mathcal{P}(Y_Q)} \frac{1}{|D_Q|} \sum_{i=1}^{|D_Q|} 1\left\{\hat{y}_i = p\left(y_i\right)\right\} \tag{9}$$

where $\mathcal{P}(Y_Q)$ denotes the set of all possible permutations of the class labels in $D_Q$. The optimal assignment is calculated using the Hungarian algorithm (W, 1955).

**Implementation details:** As in recent works (S. Vaze, 2022; Z. Sheng, 2023; W. Xin, 2023), we adopt ViT-B-16 (A. Kolesnikov & Zhai, 2021) as the feature extraction backbone, that has been pre-trained with DINO (C. Mathilde, 2021) on Imagenet. A three-layer multi-layer perceptron (MLP) serves as the projection head, generating a 768-d feature vector as the output. Fine-tuning the ViT backbone is limited to the last block. For all the experiments, a batch size of 128 is utilized. The reported results have been obtained by training for 50 and 100 epochs on coarse-grained and fine-grained datasets respectively. We use SGD with a momentum of 0.9, initial learning rate of 0.01 gradually annealed using a cosine scheduler. The active learning budget is kept to be triple the number of labeled classes. All our experiments utilize a single NVIDIA RTX A5000 GPU. We use 0.0001 as the constant threshold for the margin-based OACD baseline. The constant threshold for uncertainty based OACD baseline is kept at 0.7. Confusing buffer size is kept at twice the size of active learning budget. The classifier alignment frequency is kept at 10. The adaptive threshold parameter (computed based on validation set) is kept at 3 for Food 101, Imagenet-100 and Stanford Cars datasets, 4 for Oxford Pets and Cifar 10. CUB-200-2011 and Cifar100 use g = 2.5 and 3.5 respectively. Additional implementation details are given in Appendix A.1.

**Experimental Results for OCD task**: Table 2, 3 report the performance of SynC on the three coarse-grained and four fine-grained datasets using the 'Strict-Hungarian' protocol. *All* denotes the overall accuracy

| Methods | Oxford Pets | | | Food101 | | | CUB-200-2011 | | | Stanford Cars | | |
|---------|-----|-----|-----|-----|-----|-----|-----|-----|-----|-----|-----|-----|
| | All | Old | New | All | Old | New | All | Old | New | All | Old | New |
| **Margin-based baseline** | _60.56_ | **81.12** | _49.75_ | _32.50_ | 54.95 | **21.05** | _46.28_ | _67.47_ | _35.67_ | _25.92_ | _44.52_ | 16.72 |
| **Uncertainty-based baseline** | _62.13_ | 62.67 | _61.84_ | 32.37 | _57.15_ | _19.73_ | _46.57_ | _71.40_ | 34.13 | 25.29 | _36.72_ | _19.62_ |
| **SynC-AL** | **63.04** | _64.37_ | **62.34** | **34.68** | 64.59 | 19.43 | **50.56** | _67.47_ | **42.08** | **26.10** | _36.72_ | **20.85** |

Table 5: Experimental results showing the baselines of OACD setting and our framework 'Sync-AL' across four fine-grained datasets.

Figure 3: All the above samples are incorrectly marked as novel by SynC. Out of these examples, the samples correctly identified as belonging to previously-seen classes by SynC-AL are in green box, whereas samples that are wrongly marked by both SynC and SynC-AL are shown in the red box.

while *Old* and *New* denote the accuracies for known and novel classes. The results of the other approaches are taken directly from (Du et al., 2023). SynC consistently outperforms the state-of-the-art approaches for all course-grained and three of the four fine-grained datasets, except Stanford Cars, making it an effective framework for OCD. Bold and underline represents the highest and second highest accuracy respectively.

**Experimental Results for OACD task**: Since there are no existing baselines for the *On-the-fly Active Category Discovery* (OACD) task, we compare the proposed SynC-AL framework with two strong *baselines, which uses full supervision in active learning module* in contrast to our extremely light-weight active learning module. Specifically, we report results on (i) Constant confidence and (Prabhu et al., 2021) (ii) Margin threshold.(Xie et al., 2022)(Gui et al., 2024) Note that the active learning module is used to detect examples from novel classes, and thus standard baselines like random sampling, etc. cannot be utilized.

The performance of the two baselines and the proposed SynC-AL are reported in Tables 4, 5 for the course-grained and fine-grained datasets respectively. We observe that the proposed framework performs favorably with respect to the two baselines. For instance, while the OACD uncertainty-based baseline attains 19.73% accuracy on new classes in Food101, its old-class accuracy plummets to 57.15%. Our SynC-AL framework not only maintains robust performance on new classes (19.43%) but also significantly enhances old-class retention (64.59%), achieving a higher overall accuracy (34.68% vs. 32.50%). This pattern persists across other datasets such as CUB-200-2011, where SynC-AL increases the overall accuracy to 50.56% while also closing the gap between old and new classes.

**Note that the two baselines uses complete supervision**. Complete supervision implies that the oracle gives the class names of the query samples predicted as belonging to novel classes. So new class nodes are only created for the genuine new classes. In the proposed light-weight SynC-AL framework, if an example from a seen class does not belong to the top-k predicted classes, it will be mistakenly considered as a novel class, leading to degraded performance. Also, unlike our framework, complete supervision requires more domain expertise. Still, the proposed light-weight SynC-AL is able to correct several misclassified samples, that were mistakenly identified as novel by SynC, but actually belong to previously seen classes as shown in Figure 4. This correction improves the accuracy of old classes, as these samples are reassigned to their correct categories. Few qualitative results are shown in Figure 3.

Figure 5 exhibits the growth of the confusion buffer while encountering samples during inference. It is important to note that the maximum size of the confusion buffer is bounded by the active learning budget, which typically constitutes a small fraction of the total number of input samples. In SynC-AL, the buffer can grow up to the full active learning budget. However, we find that even a reasonably small confusion buffer can significantly enhance performance.It can be observed in figure 5 that the growth of the confusion buffer slows down as the model processes more samples.

Table 6: Performance (accuracy %) of our framework SynC(w/o Classifier Alignment) with varying values of g used for adaptive class-specific threshold computation.

| g | Imagenet-100 | | | Stanford Cars | | |
|---|---|---|---|---|---|---|
| | All | Old | New | All | Old | New |
| 3 | 42.96 | 89.45 | 19.60 | 23.36 | 30.50 | 19.82 |
| 3.5 | 43.76 | 89.85 | 20.59 | 20.69 | 25.37 | 18.38 |
| 4 | 43.81 | 87.70 | 21.75 | 20.68 | 28.87 | 16.62 |

Table 7: Effect of frequency of classifier update. Performance (accuracy %) when classifiers are updated after a certain number of novel classes are discovered.

| Classifier Alignment | CIFAR 100 | | | CUB-200-2011 | | |
|---|---|---|---|---|---|---|
| | All | Old | New | All | Old | New |
| 10 | 56.13 | 68.43 | 31.54 | 49.44 | 49.07 | 49.63 |
| 20 | 55.51 | 66.83 | 32.87 | 45.33 | 54.27 | 40.85 |
| 30 | 55.64 | 66.95 | 33.02 | 43.12 | 50.33 | 39.51 |

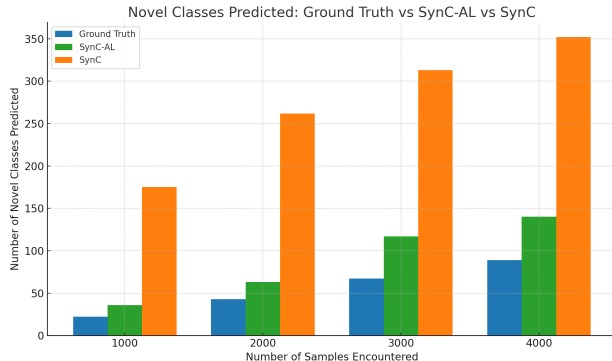

Figure 4: Barplot showing how estimated number of novel classes improves using SynC-AL.

**Additional Analysis:** Here, we conduct additional analysis of the different parameter choices.

**1) Effect of Adaptive Threshold Parameter:** Table 6 shows how different values of the hyperparameter $g$ in the adaptive threshold step (Eqn. (5)) impacts the balance between old and new class performance, with smaller $g$ slightly favoring old-class accuracy (e.g., 89.45% for $g=3$ on Imagenet-100) and larger $g$ slightly improving new-class accuracy (e.g., 21.75% for $g=4$). But the performance is quite robust with different values of $g$.

Table 8: Performance (accuracy %) of our framework SynC with different text encoder, namely SBERT and CLIP

| Method | CUB-200-2011 | | | Stanford Cars | | |
|---|---|---|---|---|---|---|
| | All | Old | New | All | Old | New |
| PHE | 36.4 | 55.8 | 27.0 | **31.3** | **61.9** | 16.8 |
| SynC(SBERT) | *45.33* | *54.27* | *40.85* | 24.6 | 34.8 | *19.55* |
| SynC(CLIP) | **52.31** | **54.73** | **51.10** | *29.7* | *45.21* | **22.02** |

Table 9: Effect of varying the number of text prompts used for generation of textual descriptions of each class.

| Number of text prompts | Oxford Pets | | |
|---|---|---|---|
| | All | Old | New |
| 2 | 60.01 | 76.25 | 51.75 |
| 4 | 61.65 | 69.46 | 57.55 |

**2) Effect of Classifier Alignment Frequency:** We observe from Table 7 that more frequent classifier alignment updates (every 10 novel classes) generally enhances performance on both CIFAR-100 and CUB-200-2011 datasets, particularly improving accuracy on old classes. On the other hand, decreasing the update frequency slightly boosts accuracy on novel classes for CIFAR-100, suggesting a trade-off between preserving old-class accuracy and enhancing novel-class recognition.

**3) Effect of different text encoders** : We conducted the same experiment on the CUB-200-2011 and Stanford Cars datasets using the same set of prompts, but with CLIP replacing SBERT for generating text embeddings as can be seen in Table 8. As expected, we observe a significant improvement in performance due to presence of the stronger text encoder.

**4) Effect of text prompts used for description generation**: The prompts are manually chosen and the performance of the framework do depend upon the number and quality of prompts. That being said, having more numbers of prompts increases the time taken for the pre-processing step. On the matter of quality of prompts, more particular and data-specific prompts perform better than generalized questions. For example, for the dataset CUB(a dataset consisting of 200 species of birds), "What does the beak of the bird look like ?" will perform better than "describe the bird." All our results for the Oxford Pets dataset are executed with 4 prompts. We then experimented by reducing the number of prompts, and a visible decrease in performance is noticeable in Table 9 thus supporting our previous insights.

**5) Ablation Study:** We analyze the contribution of each component of our framework (Table 10). Starting from a simple supervised contrastive baseline ($\mathcal{L}^s$), we progressively introduce textual contrastive loss ($\mathcal{L}^t$), prototypical contrastive loss ($\mathcal{L}^p$) and classifier alignment. Finally, we incorporate lightweight active learning (SynC-AL) with and without a confusion buffer. This analysis clearly demonstrates that each additional component incrementally enhances overall performance, validating their importance.

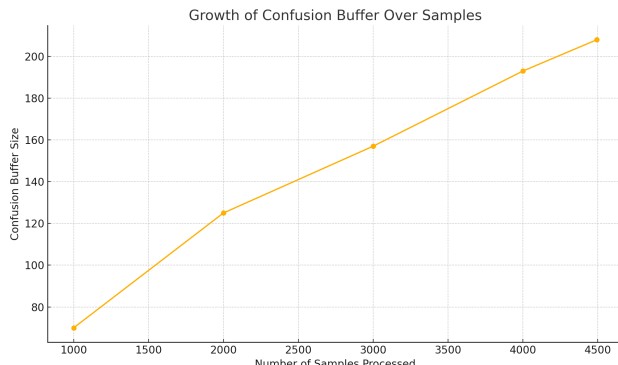

Figure 5: Graph illustrating the change in the size of the confusion buffer as the model encounters samples during inference.

| Method | $L_s$ | $L_t$ | $L_p$ | ClassifierAlign. | AL | Conf.Buffer | CIFAR-10 | | | CUB-200-2011 | | | Stanford Cars | | |
|---|---|---|---|---|---|---|---|---|---|---|---|---|---|---|---|
| | | | | | | | All | Old | New | All | Old | New | All | Old | New |
| Baseline ($L^s$ only) | ✓ | × | × | × | × | × | 37.04 | 67.24 | 21.94 | 30.22 | 46.53 | 22.04 | 21.09 | 31.05 | 16.15 |
| $L^s + L^t$ only | ✓ | ✓ | × | × | × | × | 38.52 | 57.04 | 29.26 | 35.65 | 56.53 | 25.18 | 21.87 | 32.43 | 16.64 |
| $L^s + L^p$ only | ✓ | × | ✓ | × | × | × | 39.06 | 34.05 | 41.57 | 33.89 | 53.33 | 24.15 | 23.2 | 35.6 | 18.55 |
| $L^s + L^t + L^p$ | ✓ | ✓ | ✓ | × | × | × | 49.85 | 39.80 | 54.88 | 37.12 | 56.67 | 27.32 | 21.92 | 27.64 | 19.09 |
| $L^s + L^t + L^p$ +Classifier Align. | ✓ | ✓ | ✓ | ✓ | × | × | 50.13 | 40.57 | 54.90 | 45.33 | 54.27 | 40.85 | 24.6 | 34.8 | 19.55 |
| SynC-AL (AL, w/o Conf. Buffer) | ✓ | ✓ | ✓ | ✓ | ✓ | × | 51.75 | 39.94 | 57.66 | 49.11 | 63.40 | 41.95 | - | - | - |
| **SynC-AL (Full model)** | ✓ | ✓ | ✓ | ✓ | ✓ | ✓ | **52.12** | **39.65** | **58.35** | **50.56** | **67.47** | **42.08** | **26.10** | **36.72** | **20.85** |

Table 10: Ablation study highlighting contributions of each component in SynC and SynC-AL frameworks.

| Methods | CUB-200-2011 | | | Estimated #Novel Class | Estimated #class | Stanford Cars | | | Estimated #Novel Class | Estimated #class |
|---|---|---|---|---|---|---|---|---|---|---|
| | All | Old | New | | | All | Old | New | | |
| SMILE (16bit) | 31.9 | 52.7 | 21.5 | 824 | 924 | 27.5 | 52.5 | 15.4 | 798 | 896 |
| PHE (16bit) | 37.6 | 57.4 | 27.6 | 218 | 318 | 31.8 | 65.4 | 15.6 | 611 | 709 |
| SMILE (32bit) | 27.3 | 52.0 | 14.97 | 2046 | 2146 | 21.9 | 46.8 | 9.9 | 2855 | 2953 |
| PHE (32bit) | 38.5 | 59.9 | 27.8 | 374 | 474 | 31.5 | 64.0 | 15.8 | 664 | 762 |
| SMILE (64bit) | 22.6 | 45.3 | 11.2 | 2810 | 2910 | 16.5 | 38.2 | 6.1 | 4690 | 4788 |
| PHE (64bit) | 38.1 | 60.1 | 27.2 | 393 | 493 | 32.1 | 66.9 | 15.3 | 819 | 917 |
| **SynC** | 45.33 | 54.27 | 40.85 | *395* | *495* | 24.60 | 34.60 | 19.55 | *427* | *525* |
| **SynC-AL** | 51.27 | 67.27 | 43.25 | **155** | **255** | 26.10 | 36.72 | 20.85 | **349** | **447** |

Table 11: Sync-AL provides a better estimate of the number of classes compared to SynC'. Correct number of classes are 200 and 196 for CUB and Stanford Cars respectively.

We hypothesize that the textual embedding space is limited for some datasets, especially in the case of Stanford Cars. The resolution needed for differentiating between "Chevrolet Corvette ZR1 2012" vs "Chevrolet Corvette Convertible 2012" is lacking in the textual embedding space provided by SBERT. Thus, it results in poor performance of our framework. We observe that the performance gain from using text embeddings on this dataset is minimal as seen in Table 10, suggesting that a stronger text encoder like CLIP might yield better results which we obtained in Table 8. The stronger encoder substantially improves accuracy on both old and new classes for Stanford Cars. Although richer class descriptions might boost performance further, we leave that exploration to future work.With CLIP, our framework SynC ranks second only to PHE (Z. Haiyang, 2024) on Stanford Cars; on every other dataset, however, it significantly outperforms PHE and all competing methods, even when using SBERT.

**6) Comparison with SOTA**: The state-of-the-art OCD approaches PHE (Z. Haiyang, 2024) and SMILE (Du et al., 2023) rely on hash codes to represent category descriptors, introducing inherent challenges associated with the *predefined length of hash codes*. This fixed-length representation limits the expressiveness of category features, particularly in fine-grained settings where nuanced feature variations are crucial for distinguishing novel classes. In contrast, our proposed framework, SynC and SynC-AL, circumvents these constraints employing a *list of prototypes* as classifiers, allowing for dynamic category representation without being constrained by a fixed bit-length encoding.

In addition, the lossy nature of hash-based representations in PHE and SMILE can potentially lead to unwanted category merging or fragmentation. Our prototype-based classification enables better estimation of novel category samples. As a result, our framework produces a more accurate count of novel categories compared to the state-of-the-art approaches. We observe from Table 11 that both PHE and SMILE significantly overestimates the number of novel classes (e.g., PHE (16-bit) predicts 611 novel classes and SMILE (16-bit) predicts 798 novel classes for Stanford Cars). In contrast, the proposed SynC framework provides a significantly more accurate estimate (395 for CUB and 427 for Stanford Cars), though there is still significant room for improvement.

Moreover, both PHE and SMILE's performance is sensitive to hash code length. As seen in Table 11, increasing the bit-length from 16-bit to 64-bit does not lead to a significant improvement in PHE performance, but results in an inflated number of estimated novel categories, highlighting the limitations of hash-based representations, especially for fine-grained category discovery.

Additionally, SynC-AL significantly outperforms SynC, benefiting from an active learning component that refines category boundaries and reduces overestimation errors. Specifically, SynC-AL estimates only 155 novel classes on CUB while the correct number of novel classes is 100, compared to 395 in SynC, indicating that the integration of active learning minimizes misclassification of ambiguous instances as novel categories. Similar trends can be observed on Stanford Cars, reinforcing the effectiveness of our framework over fixed-length hash-based methods.

**Limitations:** One potential limitation of our framework is the reliance on LLM for obtaining the class representation. This might be a challenge if the class names are uncommon and may not have been present in LLM training. Another potential limitation arises when the generated text descriptions are highly similar to each other, as observed in the case of the Stanford Cars dataset, where it's inherently difficult to describe different car models distinctly. Finally, if new classes gets predicted to old classes, then it will not be identified by our active learning framework.

## 6 Conclusion

In this work, we proposed a novel framework, SynC to address the challenging and realistic OCD task. We proposed a LLM-assisted unique semantic representation for learning the image mappings in addition to a periodic classifier update module for this task. Additionally, we also propose an extended version, SynC-AL which incorporates lightweight active learning, allowing minimal human intervention to correct errors and improve performance. Our proposed SynC and SynC-AL achieved state-of-the-art performance across several benchmark datasets, significantly outperforming existing methodologies.

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

| Datasets | Text prompts used for description generation | |
| --- | --- | --- |
| | Number | Prompts |
| **CIFAR10** | 3 | Describe what an {} looks like
Describe a(n) {}:
What are the identifying characteristics of a(n) {}? |
| **CIFAR100** | 4 | Describe a photo of a(n) {}:
Describe what a(n) {} looks like:
Describe a(n) {}:
What are the identifying characteristics of a(n) {}? |
| **ImageNet-100** | 5 | Describe what a(n) {} looks like
A caption of an image of a(n) {}:
How can you identify a(n) {}?
What does a(n) look like?
Describe an image from the internet of a(n) {} |
| **CUB-200-2011** | 3 | Describe the bird {} :
Describe what the bird {} looks like.
What are the identifying characteristics of the bird,a(n) {}? |
| **Stanford Cars** | 9 | How can you identify a(n) {}?
Description of a(n) {}, a type of car
A caption of a photo of a(n) {}:
What are the primary characteristics of a(n) {}?
Description of the exterior of a(n) {}
What are the identifying characteristics of a(n) {}, a type of car?
Describe an image from the internet of a(n) {}
What does a(n) {} look like?
Describe what a(n) {}, a type of car looks like |
| **Food 101** | 4 | Describe what a(n) {} looks like
Describe the color and shape of a(n) {} :
What does a(n) {} taste like?
what is the cuisine of a(n) {} ? |
| **Oxford Pets** | 4 | Describe what a(n) {} looks like
Describe the key physical characteristics a(n) {}
for example:(size, coat type , colors, and distinctive features)
What is the typical temperament and interaction style a(n) {} ?
What is the origin or history of a(n) {} ? |

Table 12: Table denoting the prompts used for text generation using LLM

# A Appendix

## A.1 Implementation Details

We used Llama-2-13b-chat-hf for the generation of our text descriptions from prompts appended with class-names. Following this, we used Sentence-BERT to generate our text embeddings using the prompts mentioned in table 12.

## A.2 Baselines pertaining to On-the-fly Active Category Discovery

**On-the-fly Active Category Discovery** (OACD) requires that each sample be inferred instantaneously, and the decision to request for human intervention must be made in real-time. Since inference occurs

one sample at a time, traditional active learning strategies such as random sampling, Query-by-Committee (QBC)(S. H. Sebastian, 1992) cannot be directly applied.

**Baseline Sampling Strategies**

### A.2.1  Constant Confidence Threshold

We employ a constant confidence threshold as the sampling strategy. Given an image sample $x_i$ with an $l_2$-normalized feature representation $z_i$, we compute its cosine similarity with the prototypical class representatives $(p^1, p^2, ..., p^L)$ of previously seen categories. The similarity score for class $c$ is defined as:

$$s_c = \cos(z_i, p^c) = \frac{z_i \cdot p^c}{\|z_i\|\|p^c\|}, \quad c \in \{1, ..., L\}. \tag{10}$$

The highest similarity score across all previously seen categories is given by:

$$s_{\max} = \max_{c \in \{1,...,L\}} s_c. \tag{11}$$

If $s_{\max}$ falls below a predefined confidence threshold $\tau$, the sample is considered a probable novel instance:

$$s_{\max} < \tau. \tag{12}$$

In such cases, we activate the active learning module to determine whether the sample belongs to an entirely new category or aligns with an existing one.

### A.2.2  Constant Margin Threshold

In addition to the constant confidence threshold, we implement a constant margin threshold as an alternative sampling strategy. Instead of relying on an absolute similarity threshold, we evaluate the relative difference between the highest and second-highest cosine similarities of the $l_2$-normalized feature representation $z_i$ with the prototypical class representatives.

Let the top two similarity scores be defined as:

$$s_{\max} = \max_{c \in \{1,...,L\}} s_c, \quad s_{2\text{nd}} = \max_{c \in \{1,...,L\} \setminus \arg\max s_c} s_c. \tag{13}$$

The margin between these two scores is computed as:

$$\Delta s = s_{\max} - s_{2\text{nd}}. \tag{14}$$

If this margin falls below a predefined threshold $\tau_m$, the sample is considered ambiguous and likely to belong to a novel category:

$$\Delta s < \tau_m. \tag{15}$$

In such cases, we trigger the active learning module to determine whether the sample requires further supervision.

### A.3  Algorithm

### A.4  Semantically similar classes exhibit similar variations in their feature distributions

To evaluate whether semantically similar classes exhibit similar variations in their feature distributions, we perform the following experiment:

---

**Algorithm 1** Light-Weight Active Learning module of SynC-AL

---

Input: Test set $X_Q$ ; prototypical class representative classifiers $P = \{p^1, p^2, ..., p^L\}$
Output: prediction
**while** $x_i \in X_Q$ **do**
    Compute normalized feature $z_i$ from $x_i$
    **for** each class $c \in |P|$ **do**
        Compute cosine similarity $sim_c$ between $z_i$ and $p^c$
        **if** $sim_c \geq \tau_c$ **then**
            Assign $x_i$ to class $c$
            $prediction$ = class $c$
        **end if**
    **end for**
    **if** $sim_c < \tau_c \forall c$ **then**
        Check against Confusing Buffer $\mathcal{M}$
        **if** Similarity with confusing sample is high **then**
            Classify $x_i$ accordingly
            $prediction$ = index of the class
        **else**
            Use Active learning budget if available
            **if** $x_i \in$ known classes **then**
                Update Confusing Buffer $\mathcal{M}$
                $prediction$ = index of the class
            **else**
                $x_i \in$ novel class. Update $P$ accordingly
                $prediction$ = last index of $P$
            **end if**
        **end if**
    **end if**
**end while**

---

First, for every pair of classes, we compute their text-embedding similarity, defined as the cosine similarity between their normalized text embeddings. This quantifies how semantically similar the two classes are.

Second, we compute a covariance-based similarity by measuring the Frobenius norm of the difference between their feature covariance matrices and converting this distance into a similarity score (e.g., using an exponential transformation).

Finally, we compute an agreement score. For each class, we check whether its nearest neighbor (based on text-embedding similarity) is among its top-3 most similar classes based on covariance similarity. The percentage of classes for which this condition holds is then reported for different datasets. For CIFAR-10, Oxford Pets, and Food-101, the agreement scores are found to be 90.0%, 78.4%, and 82.2%, respectively, supporting the underlying assumption.

For example, in the OxfordPets dataset, "American pit bull terrier" is a dog breed whose text embeddings are close to that of "Staffordshire bull terrier", and its top-3 most similar classes based on covariance similarity are "Staffordshire bull terrier", "boxer", and "pug".

### A.5 Few experimental results

Here we have provided our full method, SynC( without textual regularization), and a comparison with other text encoders and SOTA PHE(Z. Haiyang, 2024), for the dataset CUB-200-2011. From Table 13, it is evident that the proposed SynC framework consistently outperforms the state-of-the-art method PHE on the fine-grained CUB dataset, even in the absence of textual guidance (i.e., when using vision-only input). Moreover, replacing SBERT with RoBERTa, whose pretraining corpus does not include image captions, still results in a notable performance gain. These findings indicate that the improvement arises from the models effective use of distributional semantics, rather than reliance on implicit visual priors within the language model.

Here we provide the results of our framework Syncs performance on another fine-grained dataset, Arachnida, a subset of iNaturalist(Van Horn et al., 2017) dataset.

| Text Encoder Variant | ALL | OLD | NEW |
|---|---|---|---|
| PHE | 36.40 | 55.80 | 27.00 |
| SynC(Vision only) | 43.17 | 72.67 | 28.39 |
| SynC(RoBERTa) | 39.30 | 55.60 | 31.13 |
| Sync(SBERT) | 45.33 | 54.27 | 40.85 |
| SynC(CLIP) | 52.31 | 54.73 | 51.10 |

Table 13: Performance across Text Encoder Variants

| Method | ALL | OLD | NEW |
|---|---|---|---|
| SMILE | 29.90 | 57.90 | 12.20 |
| PHE | 37.00 | 75.70 | 12.60 |
| SynC | 43.24 | 70.97 | 30.87 |

Table 14: Performance of our framework SynC on 'Arachnida'.

The results in Table 14 show that our SynC framework surpasses the current state-of-the-art method, PHE. The dataset Arachnida includes fine-grained class names such as Loxosceles reclusa, Mastigoproctus giganteus, and Menemerus bivittatus, which are unlikely to be present in the pretraining corpus of SentenceBERT. This suggests that the observed performance gains are not a result of any incidental visual-language alignment in the text encoder. For all experiments, the adaptive threshold parameter $g$ is set to 3.5, and we use two prompts per known class to generate textual descriptions.

