# OpenReview forum: "Language-assisted Feature Representation and Lightweight Active Learning For On-the-Fly Category Discovery"
_TMLR — Accepted by TMLR_

### Review · Reviewer_EQNJ · 2025-06-15

**Summary Of Contributions:**

This work proposes an integrated framework, SynC and SynC-AL, as strong baselines to address the OCD problem.

Specifically, it encompasses self supervised contrastive learning to prevent corruption of the representation, image-text contrastive learning to maintain representation generalizability, and prototypical learning to learn visual semantic structures.

Besides, it utilizes class-specific statistical threshold as adaptive margin as the open-set classifier and novel-class classifier.

It proposes TopK comparison as cheap active learning query for test-time learning.

In contrast to previous coding based OCD methods, it can achieve SOTA performance on most of datasets.

**Audience:**

Yes

**Claims And Evidence:**

Yes

**Requested Changes:**

Provide more in-depth investigations into the language utility: (1) How does language embedding space boost the performance? Is there any potential information leakage? (2) To what extent can linguistic representation benefit the semantic learning? (3) Are there any limitations on language representation? It would be appreciated if you could include an additional breakdown of ablations on the StanfordCars dataset for examination.


Highlighting the difference between this work and conventional zero-shot learning is necessary, since their paradigms are highly homogeneous.


Lack discussions on related works using prompt augmentation for category semantics [1,2,3].
[1] Chils: Zero-shot image classification with hierarchical label sets
[2] What does a platypus look like? generating customized prompts for zero-shot image classification
[3] S3A: Towards Realistic Zero-Shot Classification via Self Structural Semantic Alignment

**Strengths And Weaknesses:**

This is the first prototypical methods that demonstrate SOTA performance on OCD problem, outperforming the previous coding based methods. I personally consider this learning paradigm is more advantageous in robustness in that code learning is highly sensitive to the code length, making it difficult to determine in practice. However, to truly prove its edge, more experiments are required (see below).

I hypothesize that the textual embedding space may be limited for some datasets, e.g., StanfordCars. It is recommended to conduct additional ablations on this dataset since it does not achieve SOTA performance on it; besides, we can try more text embedding models to gain deeper insights, e.g., how the performance will improve w/ CLIP textual encoder and degrade w/ weaker textual models. Meanwhile, how the performance will change w.r.t. the output dimensionality of the 3-layer projector? To prove that its relatively low sensitivity w.r.t. the previous coding based methods can be contributions.

The technical novelty is limited as most of its components are existing practices.


Novelty: existing problem, problem of limited real-world scenarios, innovative combinations in proposed methods
Significance: concerns on generalizability on more datasets, good improvements on some datasets, medium or low future potential
Technical: Sound, Correct, Reasonable, Appropriate

---

> ### Author Response · Authors · 2025-07-12
> **Ablation on Stanford Cars and other experimental results**
>
> We agree with the hypothesis that the textual embedding space is limited for some datasets, especially in the case of Stanford Cars, as we have mentioned in the limitations section of our paper. Here we provide the ablation on the Stanford Cars dataset.
>
> | Method                                         | ALL  | OLD  | NEW  |
> |------------------------------------------------|------|------|------|
> | **Baseline ($L^s$ only)**                         | 21.09 | 31.05 | 16.15 |
> | **$L^s$&nbsp;+&nbsp;$L^t$ only (textual descriptions)** | 21.87 | 32.43 | 16.64 |
> | **$L^s$&nbsp;+&nbsp;$L^p$ only**                     | 23.20 | 35.60 | 18.55 |
> | **$L^s$&nbsp;+&nbsp;$L^t$&nbsp;+&nbsp;$L^p$**           | 21.92 | 27.64 | 19.09 |
> | **$L^s$&nbsp;+&nbsp;$L^t$&nbsp;+&nbsp;$L^p$&nbsp;+ Classifier Align.** | 24.60 | 34.80 | 19.55 |
>
>
> We observe that the performance gain from using text embeddings on this dataset is minimal, suggesting that a stronger text encoder might yield better results. To verify this, we replace SBERT with CLIP as the text encoder and report the outcomes in the following table. The stronger encoder substantially improves accuracy on both old and new classes for Stanford Cars. Although richer class descriptions might boost performance further, we leave that exploration to future work. With CLIP, our framework ranks second only to PHE on Stanford Cars; on every other dataset, however, it significantly outperforms PHE and all competing methods, even when using SBERT.
>
> As suggested, we conducted the same experiment on the CUB-200-2011 and Stanford Cars datasets using the same set of prompts, but with CLIP replacing SBERT for generating text embeddings. As expected, we observe an improvement in performance due to the stronger text encoder. The results improve as follows:
> | **Method**      | &nbsp;**CUB 200-2011**&nbsp; &nbsp;|   |   | &nbsp;**Stanford Cars**&nbsp;&nbsp; |   |   |
> |                 | **ALL** | **OLD** | **NEW** | **ALL** | **OLD** | **NEW** |
> |-----------------|:------:|:------:|:------:|:------:|:------:|:------:|
> | PHE             | 36.40  | 55.80  | 27.00  | 31.30  | 61.90  | 16.80 |
> | SynC&nbsp;(SBERT)| 45.33  | 54.27  | 40.85  | 24.60  | 34.80  | 19.55 |
> | SynC&nbsp;(CLIP) | 52.31  | 54.73  | 51.10  | 29.70  | 45.21  | 22.02 |
>
> The output dimensionality of the 3-layer perceptron is set to 768, which must match the dimensionality of the textual embeddings (also 768). This is necessary because we use a text-supervised contrastive loss that relies on the dot product between the image features and the textual embeddings.
>
> The major contributions of this work are as follows:
> (i) We show that language supervision can be successfully leveraged for the OCD task, even when the class names of the novel categories are unknown.
> (ii) We propose a test-time classifier update to transfer knowledge between old and novel classes, thereby improving discriminability.
> (iii) To the best of our knowledge, we are the first to propose the realistic On-the-fly Active Category Discovery (OACD) setting, and we also establish a strong novel baseline, namely SynC-AL, which achieves state-of-the-art performance across multiple benchmarks.
>
> The language embeddings essentially act as an injection of additional information; thus, it is expected to boost the performance. In Table 8, the second row exhibits that the addition of the textual representatives in the framework improves the performance undoubtedly, especially for the fine-grained datasets like CUB 200-2011.
> The main model architecture that we use is the same as that used by existing methods [1][2]. Language supervision is applied only to the known classes, and the observed improvement stems from effectively leveraging the semantic information associated with these known classes. For the unseen classes, no prior information is assumed, thereby eliminating any possibility of information leakage.
> Even within the active learning framework, we do not assume that the class names of the unseen classes are provided by the oracle.
> [1]Du, Ruoyi, et al. "On-the-fly category discovery." Proceedings of the IEEE/CVF Conference on Computer Vision and Pattern Recognition. 2023.
> [2]Zheng, Haiyang, et al. "Prototypical Hash Encoding for On-the-Fly Fine-Grained Category Discovery."Proceedings of Advances in Neural Information Processing Systems (NeurIPS)(2024).
>
> In Table 8, adding textual representatives yields a 1.48% gain on CIFAR-10, but a significantly larger 5.43% boost on the fine-grained CUB-200-2011 benchmark. This discrepancy underscores the value of language supervision in scenarios where inter-class visual differences are subtle, such as distinguishing among 200 visually similar bird species. In such cases, textual information provides critical discriminative cues that purely visual features struggle to capture.

---

> > ### Comment · Reviewer_EQNJ · 2025-07-15
> >
> > My primary concern is that whether there is any information leakage when using the textual space of LM as a regularizer. My current view tends to be yes. The problem is that the SentenceBert has been trained on SNLI’s premise sentences [1] which includes abundant image captions from Flickr30K [2]. This may lead to a visually semantic meaningful embedding space which boosts the performance under the hook. The same story happens to the CLIP results. CLIP's embedding space is more semantically structured and thus leads to enhancement. I suggest that the authors better situate this paper within the vision-language domain, in addition to the pure vision OCD problem.
> >
> >
> >
> > [1] A large annotated corpus for learning natural language inference
> > [2] Guiding Long-Short Term Memory for Image Caption Generation

---

> > ### Comment · Reviewer_EQNJ · 2025-07-15
> >
> > Besides, in practice, there are many domain-specific scenarios similar to StanfordCars, which cannot benefit from a general-purpose language-based encoder.

---

> > > ### Comment · Reviewer_EQNJ · 2025-07-15
> > >
> > > Notice that I hold a somehow **contradictory** view of reviewer MKmk on that this method or OCD is not required by the community. Instead, there should be a shift from general category benchmarks to domain-specific ones, on which many updated literatures demonstrate that so-called current state-of-the-art open-- or close-- sourced VLLMs fail at fine-grained classification, let alone the category discovery tasks [1,2,3]. Therefore, I would further appreciate if the authors could include comparison results on some fine-grained benchmarks, e.g., Arachnida, Animalia, Food101, Mollusca, as in the paper [4]. This can also stress test the proposed method and strengthen the justification of leveraging the pre-trained LM embedding space.
> > >
> > >
> > > [1] Advancing Fine-Grained Visual Understanding with Multi-Scale Alignment in Multi-Modal Models
> > > [2] Why are Visually-Grounded Language Models Bad at Image Classification
> > > [3] Vision LLMs Are Bad at Hierarchical Visual Understanding, and LLMs Are the Bottleneck
> > > [4] Prototypical Hash Encoding for On-the-Fly Fine-Grained Category Discovery

---

> > > > ### Comment · Reviewer_EQNJ · 2025-07-15
> > > >
> > > > I note that these similar plausible opinions are quite common among foundation model researchers. I would further suggest the author clarify these weaknesses of the current foundation models in the related work in appendix to solidify the foundation of this paper.

---

> > > > > ### Author Response · Authors · 2025-07-16
> > > > >
> > > > > We will add a section mentioning the weaknesses of the current foundation models in the appendix section of our paper.

---

> > > > ### Author Response · Authors · 2025-07-16
> > > >
> > > > We appreciate that the reviewer acknowledges the need for the ‘On-the-fly Category Discovery’ problem statement.
> > > > PHE [1] focuses solely on solving OCD for fine-grained datasets, while our framework SynC aims to provide a more generalized solution that works for both generic as well as fine-grained datasets. Thus, for a balanced evaluation, we have reported results for three coarse-grained (Table 2 ) and four fine-grained datasets, namely Stanford Cars, CUB 200-2011, Food 101, and Oxford Pets, in Table 3. We observe that, except for Stanford Cars, the proposed framework outperforms the state-of-the-art methods on all of them significantly, justifying its effectiveness.
> > > > [1]Prototypical Hash Encoding for On-the-Fly Fine-Grained Category Discovery

---

> > > > > ### Comment · Reviewer_EQNJ · 2025-07-16
> > > > >
> > > > > Thanks for your clarification. I am aware that the focus of the PHE paper is fine-grained categorization. However, I am curious about how much gain the textual space regularization offers, as I am concerned about the potential semantic information leakage. Could you please also provide the results of your full method without textual regularization or your results on the more specialized benchmarks mentioned above for justification? Here, the reason is that these category names are unlikely to appear in the pertaining corpus of SentenceBert and thus are fairer for comparisons. Showing the result on 1-2 benchmark will be sufficiently persuasive.

---

> > > > > > ### Author Response · Authors · 2025-07-20
> > > > > >
> > > > > > Here we have provided our full method, SynC( without textual regularization), and a comparison with other text encoders and SOTA PHE, for the dataset CUB.
> > > > > > | Text encoder variant |  ALL  |  OLD  |  NEW  |
> > > > > > |----------------------|:-----:|:-----:|:-----:|
> > > > > > | PHE                  | 36.40 | 55.80 | 27.00 |
> > > > > > | **Vision only**          | **43.17** | **72.67** | **28.39** |
> > > > > > | RoBERTa              | 39.30 | 55.60 | 31.13 |
> > > > > > | SBERT                | 45.33 | 54.27 | 40.85 |
> > > > > > | CLIP                 | 52.31 | 54.73 | 51.10 |
> > > > > >
> > > > > > From the table, we observe that even after completely removing textual guidance (i.e., using vision-only input), the proposed framework outperforms PHE, the state-of-the-art method on the CUB dataset, which is fine-grained in nature. Furthermore, when SBERT is replaced with RoBERTa—whose pretraining does not involve image captions—we still observe a significant improvement. This confirms that the performance gain stems from the model’s ability to leverage distributional semantics, rather than any implicit visual supervision embedded in the language model.
> > > > > >
> > > > > > We’re actively running experiments on the fine‑grained iNaturalist subset (Arachnida) and expect to deliver results within the deadline. The full iNaturalist 2017 archive is sizable (~200 GB), so both the initial download and one-time preprocessing steps, aka text‑description generation and embedding extraction, add significant compute time. Our pipeline is already running, and we’ll report the numbers as soon as possible.

---

> > > > > > ### Author Response · Authors · 2025-07-20
> > > > > >
> > > > > > Here we provide the results of our framework Sync’s performance on another fine-grained dataset, ‘Arachnida’.
> > > > > > | Text encoder variant |  ALL  |  OLD  |  NEW  |
> > > > > > |----------------------|:-----:|:-----:|:-----:|
> > > > > > | SMILE                | 29.90 | 57.90 | 12.20 |
> > > > > > | PHE                  | 37.00 | 75.70 | 12.60 |
> > > > > > | **SynC**                 | **43.24** | **70.97** | **30.87** |
> > > > > >
> > > > > > The table demonstrates that our SynC framework outperforms the current state‑of‑the‑art method, PHE. The class names present in the dataset ‘Arachnida’, namely *'Loxosceles reclusa', 'Mastigoproctus giganteus', 'Menemerus bivittatus',* etc,  are unlikely to appear in the pretraining corpus of SentenceBert. Hence, the performance improvement is not attributable to any implicit visual information encoded in the language model.
> > > > > > Owing to time constraints, we have not yet completed experiments on the remaining iNaturalist subsets—Animalia, Mollusca, and Fungi—but those results will be included in the final version of the paper.

---

> > > > > > > ### Comment · Reviewer_EQNJ · 2025-07-21
> > > > > > >
> > > > > > > Is there any dataset-specific hyperparameter tuning here?

---

> > > > > > > > ### Comment · Reviewer_EQNJ · 2025-07-22
> > > > > > > >
> > > > > > > > My concerns have been largely addressed. Thanks for all your efforts.

---

> > > > > > > > ### Author Response · Authors · 2025-07-22
> > > > > > > >
> > > > > > > > The hyperparameters that are used for this dataset are as follows:
> > > > > > > > The adaptive threshold parameter g is set at 3.5, and we use 2 prompts to generate the text descriptions for the known classes. The remaining hyperparameters stay the same. We have not done any extra dataset-specific hyperparameter tuning here.

---

> ### Author Response · Authors · 2025-07-12
>
> The difference between **Zero-Shot Learning (ZSL)** and **On-the-fly Category Discovery (OCD)** is manifold.
> Firstly, in ZSL, all target classes are known in advance; only the labels for the unseen set are missing. In contrast, OCD deals with novel classes that are not predefined—the model must discover how many such classes exist and determine which images belong together.
> Secondly, ZSL relies on external semantics (e.g., attributes or text prompts) to recognize unseen classes. In OCD, textual descriptions are available only for the seen classes.
> In short, *ZSL extends recognition* to a known but unlabeled set via semantic matching, whereas on-the-fly discovery *dynamically discovers new categories* without prior semantic definitions.
>
> We thank the reviewer for pointing out these **relevant references**. We will include a discussion of them in the main paper.
> [1] CHiLS improves CLIP’s zero-shot accuracy on coarse classes by expanding each superclass into a set of finer hierarchical subclasses, sourced from taxonomies or generated via GPT-3, and then mapping subclass predictions back to their original superclasses through a simple reweighting of superclass probabilities. This work demonstrates that leveraging predefined hierarchical label structures can substantially boost zero-shot accuracy within a fixed taxonomy. However, it operates under the assumption that the full class hierarchy is known in advance and does not address settings involving truly novel categories.
> [2] This paper, which we have already cited in Section 3.1.1, uses large language models to automatically generate customized, class-specific natural language prompts—conditioned on class names, attributes, and contextual cues, instead of relying on fixed template phrases. These tailored prompts, when used with vision-language models such as CLIP, lead to significantly improved zero-shot classification accuracy by capturing richer semantic nuances for each category.
> [3] S3A addresses a more realistic zero-shot setting in which neither labeled examples nor ideal vocabularies are available. It proposes a Cluster–Vote–Prompt–Realign pipeline, which extracts “structural semantics” from unlabeled images by clustering them, voting on candidate labels from a broad vocabulary, refining prompts with a language model, and realigning prototypes for pseudo-supervision. The model then self-trains the CLIP image encoder using both individual and structural semantic alignment in a teacher–student framework. However, S3A assumes access to all unlabeled data upfront, whereas in the OCD setting, samples arrive in a streaming manner and inference must be performed on-the-fly, making the two settings fundamentally different.

---

> > ### Comment · Reviewer_EQNJ · 2025-07-15
> >
> > This concern has been addressed as long as include the additional discussions in appendix.

---

> > > ### Author Response · Authors · 2025-07-16
> > >
> > > As suggested, we will add the additional suggestions in the appendix.

---

### Review · Reviewer_HWe6 · 2025-06-16

**Summary Of Contributions:**

This paper explores On-the-Fly Category Discovery using LLM embeddings and active learning to address the uncertainty problem. They showed the framework applied in online learning, which is practical. The main losses are compromised by contrast losses using the embedded samples and prototypes of classes, as well as the LLM-assistant prototypes. The algorithm is almost clearly explained, but some issues need to be clarified. The experiments are thoroughly conducted with various aspects of the proposed algorithm. However, some issues need clarification. The contribution of this paper appears moderate, and I recommend identifying the main contributor to the improved performance. The LLM embedding or active learning? The LLM embeddings appear to be most prominent in the ablation study. It can be stated more impressively.

**Audience:**

Yes

**Broader Impact Concerns:**

This study can impact the vision learner in finding a new class. It is crucial in the auto-driving or other vision-based alert systems.

**Claims And Evidence:**

No

**Requested Changes:**

$\bullet$ The simulation or learning can check the following claim:
1) This approach assumes that semantically similar classes will exhibit similar variations in their feature distributions (for the novel classes, we use the covariance matrix, the mean of which is most similar. This assumption looks strong. Therefore, some validations are required.
2) The size of the confusion buffer in active learning can be recorded in the learning phase. Please draw the graph illustrating the size of the confusion buffer.

$\bullet$ I have a question regarding formula (6); specifically, the right-hand side (RHS) of formula (6) has the index of $c$. However, there is no validated index $c$ on the left-hand side. I don’t know the dependency of $c$ on $\tau_c$.

$\bullet$ I have a question about the section on Test-time update of the classifiers. At test time, the update of a classifier is performed using the cross-entropy method, which applies to the DNN classifier. However, it is not consistent with the classifier in Section 3.2.1 at first glance. Please clarify this issue.

**Strengths And Weaknesses:**

Strengths:

Online type learning, combined with active learning, is a smart strategy that shows better performance. The most prominent improvement is related to LLM's embeddings, which are aligned with the development of powerful LLM models.

Weakness:
The paper does not reveal the whole picture of learning. The dimension of S-BERT is not emphasized, and some essential parts clarified are missing.

---

> ### Author Response · Authors · 2025-07-11
> **Clarifications and few experimental results**
>
> We thank the reviewers for their insightful feedback.
> 1. The major contributions of this work are as follows:
> (i) We show that language supervision can be successfully leveraged for the OCD task, even when the class names of the novel categories are unknown.
> (ii) We propose a test-time classifier update to transfer knowledge between old and novel classes, thereby improving discriminability.
> (iii) To the best of our knowledge, we are the first to propose the realistic On-the-fly Active Category Discovery (OACD) setting, and we also establish a strong novel baseline, namely SynC-AL, which achieves state-of-the-art performance across multiple benchmarks.
> 2. The dimension of the text embeddings obtained from S-BERT is 768. The MLP projection head has both input and output dimensions of 768. There are a total of 3 layers, and the hidden layer dimension is 2048. Each of the prototypical class representatives is a 768-dimensional vector.
> 3. To evaluate whether semantically similar classes exhibit similar variations in their feature distributions, we perform the following experiment:
> First, for every pair of classes, we compute their text-embedding similarity, defined as the cosine similarity between their normalized text embeddings. This quantifies how semantically similar the two classes are.
> Second, we compute a covariance-based similarity by measuring the Frobenius norm of the difference between their feature covariance matrices and converting this distance into a similarity score (e.g., using an exponential transformation).
> Finally, we compute an agreement score. For each class, we check whether its nearest neighbor (based on text-embedding similarity) is among its top-3 most similar classes based on covariance similarity. The percentage of classes for which this condition holds is then reported for different datasets. For CIFAR-10, Oxford Pets, and Food-101, the agreement scores are found to be 90.0%, 78.4%, and 82.2%, respectively, supporting the underlying assumption.
> For example, in the OxfordPets dataset, “American pit bull terrier” is a dog breed whose text embeddings are close to that of “Staffordshire bull terrier”, and its top-3 most similar classes based on covariance similarity are “Staffordshire bull terrier”, “boxer”, and “pug”.
> 4. For the CUB-200-2011 dataset, the size of the confusion buffer evolves as follows: after processing 1,000 samples, the buffer size is 70. Subsequently, after 2,000, 3,000, 4,000, and 4,494 (the total number of samples), the buffer size increases to 125, 157, 193, and 208, respectively. As observed, the growth of the confusion buffer slows down as the model processes more samples.
> It is important to note that the maximum size of the confusion buffer is bounded by the active learning budget, which typically constitutes a small fraction of the total number of input samples. In SynC-AL, the buffer can grow up to the full active learning budget. However, we find that even a reasonably small confusion buffer can significantly enhance performance.
> 5. The index c in eqn (6) denotes that for every previously seen class c {where c=1,...,L+m }, the cosine similarity between its corresponding prototypical classifier and image feature is compared against its class-specific threshold tau_c.
> 6. In Section 3.2.1, we mention that adaptive thresholding is applied to the cosine similarity between the dictionary of prototypical class representatives (from previously seen classes) and the normalized feature of the current image for classification. However, as we begin to encounter samples from novel classes, it becomes essential to update this dictionary of prototypical class representatives—i.e., to perform a test-time update of the classifiers.
> To achieve this, we employ a bias-free fully connected layer with L₂ normalization, having a 768-dimensional input and an output dimension equal to the number of prototypical class representatives. These representatives serve as the weights of the layer, which are updated during backpropagation using the cross-entropy loss.

---

> > ### Comment · Reviewer_HWe6 · 2025-07-15
> > **Reply**
> >
> > Thanks for your detailed responses. Many issues are resolved. However, I cannot avoid the feeling that there are many heuristics, especially in the similarity of covariance. In some datasets, there can be different behaviors of covariance despite a similar mean vector. Therefore, the clarification of this problem would be cautious.

---

> > > ### Author Response · Authors · 2025-07-16
> > >
> > > Thank you for the comment. We agree with the reviewer that covariance patterns vary from one dataset to another. Since the class names of the novel categories are unknown throughout the inference, in our framework, we adopt a simple rule: we consider the covariance of the class with the closest mean as an approximate representation of the variations of the novel class. It helps to generate augmented data for the new classes (and thus a reasonable approximation is sufficient), which are eventually used for classifier alignment. We showed, using extensive evaluation (in the last rebuttal), that this is a good approximation and empirically works well to significantly boost the performance.

---

> > > > ### Author Response · Authors · 2025-07-20
> > > > **A gentle reminder**
> > > >
> > > > We respectfully note that we have addressed all questions raised so far. If any additional clarification is needed, we will be happy to provide it within the stipulated deadline.

---

### Review · Reviewer_MKmk · 2025-06-29

**Summary Of Contributions:**

The paper proposes an image classification method, where the number and nature of classes are unknown. It is trained on a dataset on a subset of known classes. During inference, images are received one-by-one. Each each image can belong to one of the known classes, or to a novel class. The goal of the method is accurately classify images, as they appear, in an online setting. The authors propose a method which combines large language model based descriptions of existing class labels, contrastive learning, and active learning to accurately classify images in this setting. The method is evaluated against certain baselines, on a widely-used image classification benchmarks.

**Audience:**

Yes

**Broader Impact Concerns:**

I do not foresee any broader impact concerns. The paper also does not have a broader impact section.

**Claims And Evidence:**

Yes

**Requested Changes:**

Please respond to the questions in the previous section. Please prioritized responding to questions / comments 4 -- 7.

**Strengths And Weaknesses:**

## Strengths
1. The paper is well written and easy to follow.
2. The method performs well on the compared datasets, against the compared baselines.
3. The experimental analysis appears to be robust, especially the ablation experiments.

## Weaknesses
1. **Formal definition of the problem and constraints:** The paper lacks a formal definition of the problem at hand. I would encourage the authors to express the assumptions and constraints of the problem at hand. For example, are we assuming that the images in novel categories are drawn from the same distributions (e.g., old classes: animal, novel classes: animals, or novel classes: cars). Similarly, the constraints must be clearly mentioned. For example, specify and describe the constraints that images cannot be stored due to storage or privacy concerns (Page 6), and other resource constraints (Page 7), while formulating the problem. The differences between Generalized, Online, and On-the-fly Category Discovery are unclear. I am not sure what the key differences between these different problems are. Finally, it is unclear what happens if labels are not of the same "level". For example, given an image of a horse, should it be categorized as a  horse or an animal?
2. **Outputs:** My understanding is that the output of the method is not a class label for novel classes. Is my understanding accurate? How are class labels for novel classes generated?
3. **Prompts:** It appears that the prompts $(v_1, \dots, v_d)$ are manually chosen. How does the number and quality of prompts impact the method?
4. **Experimental protocols:**
    1. Why are the baseline methods in Table 2 and 3 different? Are the baselines latest?
    2. It is unclear if the gains in some cases are significant. I recommend that the authors run each experiment with 3 different random seeds, and different samples, and numbers of known and unknown classes, to ascertain if the performance gains are significant. The means and standard deviations across these runs should be reported.
    3. I am not sure what the strict Hungarian protocol entails.
5. **Prior Work (minor):**
    1. The idea of using LLMs for zero- (or few-) shot classification is not new. Some studies such as [1] have already used similar concepts before.
    2. I am not sure what the downsides of having "hash codes of predefined lengths" are?
6. **How present/future-proof are these methods?** Perhaps, the biggest issue I have is if the proposed methods perform on par with Large Vision Language Models. How do LVLMs such as ChatGPT or Claude perform on this task? Why do we need such a method?
7. **Results:** Why is performance on the Stanford Cars dataset poor?
8. **Hyper-parameters:** How are hyper-parameters for the method chosen?


## References
1. Gao, Chufan, et al. "Classifying unstructured clinical notes via automatic weak supervision." Machine Learning for Healthcare Conference. PMLR, 2022.

---

> ### Author Response · Authors · 2025-07-13
> **Clarifications**
>
> [1]  Let  $D_L$ =$\\{(x_i, y_i)\\}_{i=1}^N ∈ X_l × Y_l $, be a labeled set of “old” classes $ y_i​∈Y_l​$, and $D_U$ = $\\{x_i\\}$ where (i=1,2,..,M) and $ x_i ∈ X_u$, be the set of unlabeled samples. $D_U$ comprises samples from both known (seen during training) and novel categories. So, $Y_l ⊂Y_u$.
> We make the following assumptions and constraints:
> [i]Shared domain assumption. All images—old and novel—come from the same overall visual distribution (e.g., all the images are photographs of animals).
> [ii]Uniform granularity. Both old and novel labels reside at the same semantic level (e.g., “horse” vs. “zebra,” not “animal” vs. “horse”).
> [iii]Privacy/storage: Raw images cannot be stored long-term; only their learned embeddings or prototypes may be kept.
> The datasets that are used for the experimentations contain labels of the same taxonomy level.
>
> **Generalized Category Discovery**: The model has access to both the labeled dataset $D_L$ and the unlabeled dataset $D_U$ at the time of training. The unlabeled dataset contains samples from previously seen as well as unseen novel classes. The model is tasked with predicting the labels of the samples in $D_U$ at test time.
> **On-the-fly Category Discovery**: The model is trained only on $D_L$. During inference, the model encounters samples from $D_U$, as they arrive one at a time in an online manner. The task is to assign each of the incoming samples to either previously encountered classes or novel, unseen categories, instantaneously.
>
> **[2]** The output of SynC is a numerical class label, even for novel classes. The numerical labels assigned to the novel classes might not be correct; hence, relabelling is done using the Hungarian algorithm (mentioned later on in this discussion).
>
> **[3]** The prompts are manually chosen, and the performance of the framework does depend upon the number and quality of prompts. That being said, having more numbers of prompts increases the time taken for the pre-processing step. Now, on the matter of quality of prompts, more particular and data-specific prompts perform better than generalized questions. For example, for the dataset CUB(a dataset consisting of 200 species of birds), *“What does the beak of the bird look like ?”* will perform better than *“describe the bird.”*
> We experimented with the number of prompts for the dataset Oxford Pets, and the results support our previous insights.
>
> | Number of prompts |  ALL  |  OLD  |  NEW  |
> |:----------------------------:|:-----:|:-----:|:-----:|
> | 2                           | 60.01 | 76.25 | 51.75 |
> | 4                           | 61.65 | 69.46 | 57.55 |
>
> All our results for the Oxford Pets dataset are executed with 4 prompts. We then experimented by reducing the number of prompts, and a visible decrease in performance is noticeable.
>
> **[4]** Experimental protocols:
>
>   [1]The paper “Prototypical Hash Encoding for On-the-Fly Fine-Grained Category Discovery” by H. Zheng, et. al. reports their methodology for the fine-grained datasets only, thus making PHE the SOTA for fine-grained datasets (namely CUB 200-2011, Stanford Cars, Food101, and Oxford Pets). So in Table 2, we use SMILE as SOTA for generic datasets (CIFAR 10, CIFAR 100, and Imagenet-100) while in Table 3, we use PHE as SOTA for fine-grained datasets.
>
> [2] As suggested by the reviewer, we have reported the results with mean and standard deviation for CIFAR10 and CIFAR100 for SynC. We will incorporate the results for SynC-AL in the final version of the paper.
>
> | Dataset  |  ALL  |  OLD  |  NEW  |
> |----------|:-----:|:-----:|:-----:|
> | CIFAR-10  | 50.43&nbsp;&plusmn;&nbsp;0.02 | 40.70&nbsp;&plusmn;&nbsp;0.01 | 54.70&nbsp;&plusmn;&nbsp;0.03 |
> | CIFAR-100 | 56.02&nbsp;&plusmn;&nbsp;0.01 | 68.30&nbsp;&plusmn;&nbsp;0.02 | 31.30&nbsp;&plusmn;&nbsp;0.02 |

---

> ### Author Response · Authors · 2025-07-13
> **Clarifications**
>
> **4.** [3] Clustering (or any unsupervised grouping) produces arbitrary cluster IDs. To report a meaningful “accuracy”,i.e., the fraction of samples whose predicted label matches the true label, we first need to optimally relabel the clusters. The Hungarian algorithm finds the best one-to-one mapping between the predicted cluster indices and the ground-truth class labels that maximizes total correct assignments. In the Strict‐Hungarian protocol, we first apply the Hungarian algorithm once to the entire query set to establish a one‐to‐one mapping between predicted cluster IDs and ground‐truth labels, ensuring each cluster is paired with exactly one label. With this unified relabeling in place, we then calculate accuracy separately for the New and Old subsets using those locked‐in predictions. By doing so, no cluster can be reused across categories, resulting in a consistent, globally optimal assignment before any subset‐level evaluation. The classic Hungarian (Kuhn–Munkres) method for the assignment problem, where we need to assign each of n workers to n jobs (one-to-one) so as to minimize the total cost. If we consider a toy example with ground-truth labels [1, 1, 2, 2], where class 1 is deemed “Old” and class 2 is “New.” Suppose the model predicts cluster assignments [2, 2, 1, 1]. To maximize accuracy, one can simply relabel cluster 2 as class 1 and cluster 1 as class 2, achieving a perfect 100 percent accuracy. The Hungarian algorithm performs this optimal one-to-one relabeling automatically, finding the assignment (1 → 2, 2 → 1) in a single step.
>
> **5.** [1]Thank you for pointing out Gao et al. (2022). We agree that large-language-model (LLM) signals have been explored for zero-/few-shot text classification, and we will cite this work in our Related Work section. Our contribution, however, targets a different modality and problem setting:
> *Domain*: Gao et al. focus on generating weak labels for unstructured clinical notes (pure NLP). In contrast, our framework operates on images and tackles the dual task of recognizing known classes and discovering entirely new visual categories, for which we do not assume that the class is known, and thus cannot use the LLM supervision.
>
> *Problem formulation*: Their goal is to assign text documents to a fixed set of classes using LLM-derived labeling functions. We address open-world, on-the-fly category discovery (OCD), where the number and identity of novel classes are unknown and must be discovered online with a limited active-learning budget.
>
> *Methodological novelty*: While we do use text embeddings, our key innovation is the integration of (i) language-guided prototypes, (ii) self-supervised visual contrast, (iii) a test-time classifier-alignment mechanism, and (iv) an active-query strategy—all within a single pipeline that updates continuously as new classes emerge.
> To our knowledge, no prior work, including Gao et al., combines these elements for open-world vision tasks. We therefore believe our work is complementary to Gao et al. rather than overlapping.
>
> [2]  Using hash codes of a fixed, pre-set length is attractive for fast retrieval, but it comes with several practical constraints, especially in an open-world setting like ours. A b-bit code can represent at most 2ᵇ distinct buckets. Short codes save memory but sacrifice discriminative power; long codes improve accuracy but inflate storage and lookup time. Once the length is chosen, this trade-off cannot be adjusted on the fly. Also, if the code length later proves inadequate, all stored embeddings must be regenerated, which will be computationally expensive and often not feasible under privacy or storage constraints.
>
> **6.** The majority of the publicly available datasets, which are used for the problem of novel class discovery and for other applications too, have been used for training these LVLMs like ChatGPT, Claude, etc. So it is expected that they will be able to perform well on these datasets. But, the problem statement of On-the-fly category discovery remains interesting because for any dataset that is not freely available currently, LVLMs will not be able to recognize the object. As an example, so many new species of plants and animals are being discovered every year, whose data are not currently available for training these models. Thus, discovering them and incorporating the new knowledge is imperative.

---

> ### Author Response · Authors · 2025-07-13
> **Clarifications**
>
> **7.** We hypothesize that the textual embedding space is limited for some datasets, especially in the case of Stanford Cars. Like we have mentioned in the limitations section of our paper, the textual embedding space lacks the resolution needed for differentiating between “Chevrolet Corvette ZR1 2012” vs “Chevrolet Corvette Convertible 2012”. Thus, it results in poor performance of our framework. Here we provide the ablation on the Stanford Cars dataset.
> | Method                                         | ALL  | OLD  | NEW  |
> |------------------------------------------------|------|------|------|
> | **Baseline ($L^s$ only)**                         | 21.09 | 31.05 | 16.15 |
> | **$L^s$&nbsp;+&nbsp;$L^t$ only (textual descriptions)** | 21.87 | 32.43 | 16.64 |
> | **$L^s$&nbsp;+&nbsp;$L^p$ only**                     | 23.20 | 35.60 | 18.55 |
> | **$L^s$&nbsp;+&nbsp;$L^t$&nbsp;+&nbsp;$L^p$**           | 21.92 | 27.64 | 19.09 |
> | **$L^s$&nbsp;+&nbsp;$L^t$&nbsp;+&nbsp;$L^p$&nbsp;+ Classifier Align.** | 24.60 | 34.80 | 19.55 |
>
> We observe that the performance gain from using text embeddings on this dataset is minimal, suggesting that a stronger text encoder might yield better results. To verify this, we replace SBERT with CLIP as the text encoder and report the outcomes in the following table. The stronger encoder substantially improves accuracy on both old and new classes for Stanford Cars. Although richer class descriptions might boost performance further, we leave that exploration to future work. With CLIP, our framework ranks second only to PHE on Stanford Cars; on every other dataset, however, it significantly outperforms PHE and all competing methods, even when using SBERT. We conducted the same experiment on the CUB-200-2011 and Stanford Cars datasets using the same set of prompts, but with CLIP replacing SBERT for generating text embeddings. As expected, we observe an improvement in performance due to the stronger text encoder. The results improve as follows:
>
> | **Method**      | &nbsp;&nbsp;&nbsp;&nbsp;**CUB 200-2011** &nbsp;&nbsp;|   |   | &nbsp;&nbsp;**Stanford Cars**&nbsp; |   |   |
> |                 | **ALL** | **OLD** | **NEW** | **ALL** | **OLD** | **NEW** |
> |-----------------|:------:|:------:|:------:|:------:|:------:|:------:|
> | PHE             | 36.40  | 55.80  | 27.00  | 31.30  | 61.90  | 16.80 |
> | SynC&nbsp;(SBERT)| 45.33  | 54.27  | 40.85  | 24.60  | 34.80  | 19.55 |
> | SynC&nbsp;(CLIP) | 52.31  | 54.73  | 51.10  | 29.70  | 45.21  | 22.02 |
>
> **8.** All hyperparameters are chosen via a lightweight grid search on a held-out validation split taken only from the old-class portion of the training data, so no novel-class information is leaked. We train for a fixed 100 epochs for fine-grained datasets and 50 epochs for generic datasets, and select the combination that maximizes the clustering accuracy(ACC) on the Old class in the validation set. The same best-found setting is then used unchanged for the official test split of that dataset.

---

> > ### Author Response · Authors · 2025-07-20
> > **A gentle reminder**
> >
> > We respectfully note that we have addressed all the questions raised so far. If any additional clarification is needed, we will be happy to provide it within the stipulated deadline.

---

### Author Response · Authors · 2025-07-12
**Request for an Extension**

We are writing to respectfully request a one-week extension for submitting our rebuttal to the recent reviews of our manuscript titled “Language-assisted Feature Representation and Lightweight Active Learning For On-the-Fly Category Discovery”. We greatly appreciate the thoughtful and constructive feedback provided by the reviewers and would like to ensure that we address their comments thoroughly and carefully.
Due to unforeseen personal commitments, we would be grateful for a short extension of one week beyond the current rebuttal deadline. We are committed to submitting a comprehensive and timely response and believe this additional time will allow us to improve the clarity and completeness of our replies.
Please let us know if this extension can be accommodated. We truly appreciate your consideration.

---

### Author Response · Authors · 2025-07-13

We thank the reviewers for the thoughtful and constructive feedback. We have addressed every comment to the best of our ability, providing detailed explanations and supporting evidence wherever relevant. If any further clarification is required, we will be happy to oblige.

---

### Decision · Action_Editor_G88D · 2025-08-25

**Recommendation:** Accept as is

**Additional Comments:**

All three reviewers are recommending to accept the paper.

**Audience:**

Yes

**Audience Explanation:**

The topic of this paper, which deals with on-the-fly category discovery, is an important and relevant problem in many ML domains.  The use of LLMs for this task is relevant as well.  This paper will definitely be of interest to many in the TMLR community.

**Claims And Evidence:**

Yes

**Claims Explanation:**

All three reviewers have indicated that the paper provides convincing evidence for its claims.  Initially, the reviewers suggested several additional experiments to be run to bolster the paper's claims; the authors provided these (among other improvements) in an updated draft, at which point all the reviewers were satisfied by the paper's claims and evidence.  The authors did a good job improving the manuscript based on the reviewer feedback, and at this point I think the paper effectively supports its claims.